# Phosphoproteomics identifies microglial Siglec-F inflammatory response during neurodegeneration

Nader Morshed[1,2] iD, William T Ralvenius[3,4] iD, Alexi Nott[3,4,5,6] iD, L Ashley Watson[3,4], Felicia H Rodriguez[7] iD, Leyla A Akay[3,4] iD, Brian A Joughin[1,2] iD, Ping-Chieh Pao[3,4] iD, Jay Penney[3,4], Lauren LaRocque[1], Diego Mastroeni[8] iD, Li-Huei Tsai[3,4,9,*] iD & Forest M White[1,2,10,**] iD

## Abstract

Alzheimer's disease (AD) is characterized by the appearance of amyloid-β plaques, neurofibrillary tangles, and inflammation in brain regions involved in memory. Using mass spectrometry, we have quantified the phosphoproteome of the CK-p25, 5XFAD, and Tau P301S mouse models of neurodegeneration. We identified a shared response involving Siglec-F which was upregulated on a subset of reactive microglia. The human paralog Siglec-8 was also upregulated on microglia in AD. Siglec-F and Siglec-8 were upregulated following microglial activation with interferon gamma (IFNγ) in BV-2 cell line and human stem cell-derived microglia models. Siglec-F overexpression activates an endocytic and pyroptotic inflammatory response in BV-2 cells, dependent on its sialic acid substrates and immunoreceptor tyrosine-based inhibition motif (ITIM) phosphorylation sites. Related human Siglecs induced a similar response in BV-2 cells. Collectively, our results point to an important role for mouse Siglec-F and human Siglec-8 in regulating microglial activation during neurodegeneration.

**Keywords** Alzheimer's disease; microglia; phosphoproteomics; Siglec-8; Siglec-F
**Subject Categories** Neuroscience; Post-translational Modifications & Proteolysis; Proteomics
**Mol Syst Biol. (2020) 16: e9819**

## Introduction

Alzheimer's disease (AD) is a neurodegenerative disease which presently affects over 5.8 million Americans and 55 million people worldwide (Gaugler *et al*, 2019). Recent evidence has identified distinct stages of AD development, from: early amyloid-β (Aβ) deposition, tau hyperphosphorylation, release of pro-inflammatory cytokines, reactive gliosis, demyelination, synaptic loss, metabolic changes, and ultimately neurodegeneration (De Strooper & Karran, 2016). The impact of gliosis on this process is emphasized by recent genome-wide association studies (GWAS) that have uncovered mutations in microglial genes, including TREM2, CD33, and INPP5D, which contribute to disease pathogenesis through phosphorylation signaling (Lambert *et al*, 2013; Colonna & Wang, 2016; Jansen *et al*, 2019; Kunkle *et al*, 2019). Biochemical studies have highlighted the important role of signaling networks involving protein kinases such as cyclin-dependent kinase 5 (Cdk5), Glycogen synthase kinase 3 beta (Gsk3β), and Protein Kinase C alpha (PKCα) in mediating pathological processes linked to Aβ and phospho-Tau (Patrick *et al*, 1999; Choi *et al*, 2014; Alfonso *et al*, 2016). However, there have been few studies of how these signaling proteins affect proteome-wide phosphorylation events, or the phosphoproteome, during neurodegeneration.

Tools to measure the phosphoproteome have been developed using mass spectrometry (MS; Ficarro *et al*, 2002; Zhang *et al*, 2005; Rigbolt & Blagoev, 2012; Riley & Coon, 2016; White & Wolf-Yadlin, 2016). However, previous studies in AD have analyzed only a portion of the phosphoproteome and overlooked critical phosphotyrosine (pTyr) sites that mediate the activity of many known kinases (Henriques et al, 2007; Tagawa *et al*, 2015; Dammer *et al*, 2015; Bai *et al*, 2020; Marttinen *et al*, 2019). We aimed to measure and compare the phosphoproteomes of different mouse models of AD in order to understand signaling changes linked to neurodegeneration. We analyzed three separate mouse models of Alzheimer's disease and neurodegeneration: (i) CK-p25 (Cruz *et al*, 2003, 2006; Fischer *et al*, 2005), a model of p25/Cdk5 activation in forebrain neurons;

1    Department of Biological Engineering, Massachusetts Institute of Technology, Cambridge, MA, USA
2    Koch Institute for Integrative Cancer Research, Massachusetts Institute of Technology, Cambridge, MA, USA
3    Picower Institute for Learning and Memory, Massachusetts Institute of Technology, Cambridge, MA, USA
4    Department of Brain and Cognitive Sciences, Massachusetts Institute of Technology, Cambridge, MA, USA
5    Department of Brain Sciences, Imperial College , London, UK
6    UK Dementia Research Institute at Imperial College London, London, UK
7    Department of Chemical and Materials Engineering, New Mexico State University, Las Cruces, NM, USA
8    ASU-Banner Neurodegenerative Disease Research Center, Tempe, AZ, USA
9    Broad Institute of MIT and Harvard, Cambridge, MA, USA
10   Center for Precision Cancer Medicine, Massachusetts Institute of Technology, Cambridge, MA, USA
    *Corresponding author. Tel: +1 6173411660; E-mail: lhtsai@mit.edu
    **Corresponding author. Tel: +1 6173240403; E-mail: fwhite@mit.edu

(2) 5XFAD (Oakley *et al*, 2006), a model of Aβ toxicity; and (ii) Tau P301S (Ghetti *et al*, 2002), a model of phospho-Tau and neurofibrillary tangles. Using a sensitive sample enrichment protocol, we quantified low-abundance pTyr peptides and global phosphoserine/phosphothreonine (pSer/pThr) peptides in the hippocampus and cortex of these mice where pathology occurs.

Downstream analysis of these datasets revealed neuronal, astrocytic, and microglial phosphorylation signaling changes associated with disease. Furthermore, we observed dysregulation of Cdk5 phosphorylation substrates in these mouse models. We found that the Δp35KI mutation, which attenuates Cdk5 overaction by blocking calpain-mediated cleavage of Cdk5 activator p35 into p25 (Lee *et al*, 2000; Seo *et al*, 2014), exerted a protective effect on the neuronal CaMKII-linked phosphoproteome in mice with 5XFAD background. Across all models, we observed an increase in phosphorylation on Siglec-F (Also known as Siglec5), a member of the CD33-related Siglec family of immunoreceptors (Angata *et al*, 2004). We found that Siglec-F was upregulated on a subset of reactive microglia in CK-p25 and 5XFAD mice. These Siglec-F$^+$ microglia appeared early on in disease progression and were spatially correlated with Aβ pathology. While mouse Siglec-F does not have a direct homolog in humans, there are several human Siglecs that share features with Siglec-F (Duan & Paulson, 2020). We identified human Siglec-8, the functionally convergent paralog of mouse Siglec-F (Tateno *et al*, 2005), upregulated on microglia in postmortem brain samples from late-onset AD patients.

To assess the functional effects of Siglec-F, we designed several *in vitro* models of Siglec activation on microglia. We found that treatment with interferon gamma (IFNγ) increased Siglec-F expression using BV-2 cells, an immortalized cell line model of mouse microglia (Henn *et al*, 2009). We similarly observed that induced pluripotent stem cell-derived microglia (iMGLs) had increased Siglec-8 expression after IFNγ treatment. We found that Siglec expression in BV-2 cells activates an endocytic and pyroptotic inflammatory response in an ITIM-dependent manner. These phenotypes were rescued by blocking sialic acid substrate binding and inhibiting signaling through Janus Kinase (JAK), Src homology region 2-containing protein tyrosine phosphatase 2 (SHP-2), and Nucleotide-binding oligomerization domain, Leucine rich Repeat and Pyrin domain containing (NLRP) inflammasome pathways. Together, these data highlight Siglec-F and Siglec-8 as new markers for activated microglia and potential targets for modulating neuroinflammation in AD.

# Results

### CK-p25 phosphoproteomics identifies activated signaling preceding neurodegeneration

To characterize signaling network changes in several mouse models of neurodegeneration, we used mass spectrometry-based phosphoproteomics to quantify the pTyr phosphoproteome, global phosphoproteome, and protein expression profiles of brain tissues from diseased and wildtype (WT) mice. We first examined CK-p25 mice which can activate p25/Cdk5 signaling in forebrain neurons in an inducible manner. By removing doxycycline from the diet of these mice, we can induce gliosis within 2 weeks as well as neuronal loss and cognitive deficits by 6 weeks (Gjoneska *et al*, 2015; Mathys

*et al*, 2017). We collected tissue from adult mice (3–4 months old) induced for 2 weeks to capture signaling changes that precede neurodegeneration. Briefly, peptides from the hippocampus, cortex, and cerebellum were labeled with isobaric tandem mass tags (TMT), enriched for pTyr peptides by immunoprecipitation (IP), and analyzed by quantitative multiplex LC-MS/MS analysis using a previously described methodology (Reddy *et al*, 2016; Dittmann *et al*, 2019) (Fig 1A, Dataset EV1). To gain insight into signaling downstream of activated Cdk5 in this model, the pTyr IP supernatant of each tissue was enriched for CDK and MAPK substrates by IP with a phospho-motif-specific antibody (Joughin *et al*, 2009). We subsequently performed a deep analysis of the global (pSer/pThr) phosphoproteome on the supernatant from this second IP of cortical tissue, through reverse-phase fractionation followed by immobilized metal affinity chromatography-based pSer/pThr enrichment. Finally, protein expression profiling was performed on IP supernatant. This strategy led to the identification and quantification of 10,523 unique peptides (10% pTyr, 70% pSer/pThr, Dataset EV1) across the hippocampus and cortex of CK-p25 and CK control mice (CK-p25: $n = 3$; CK: $n = 3$). In agreement with the known forebrain-specific expression of p25 in this model, no significant changes (Fold Change (FC)$> 1.25$, $P < 1e$-2) were detected in protein phosphorylation in cerebellum samples from control and CK-p25 mice (Dataset EV1).

Among peptides from the hippocampus and cortex, 311 were upregulated and 110 were downregulated in CK-p25 mice relative to CK controls (Fig EV1A and B). Among the upregulated pTyr peptides there was a set of sites that map to proteins predicted to be primarily expressed in microglia and astrocytes in the hippocampus and cortex of CK-p25 mice (Fig 1B and Fig EV1C–E). In addition to pTyr changes, we observed a larger number of upregulated pSer/pThr sites (Fig EV1A and B). Motif analysis of the global pSer/pThr dataset found a -pS/T-P- motif with lysine in the $+ 2$ and $+ 3$ positions enriched in the upregulated set of peptides (Fig EV1F), which matches the reported substrate motif of CDKs (Beaudette *et al*, 1993; Songyang *et al*, 1994). Among the upregulated pTyr and pSer/pThr phosphosites, we found several kinases, receptors, and transcription factors, including Stat3 pY705 (FC = 3.8; $P = 7.0e$-5), Cdk1/2/3 pY15 (FC = 3.7; $P = 3.1e$-3), Galk2 pS208 (FC = 2.5; $P = 1.4e$-10), Brsk2 pS456, pT460, pS468 (FC = 1.6; $P = 1.7e$-5), Src pY535 (FC = 1.45; $P = 5.4e$-3), Lyn pY316 (FC = 1.7; $P = 1.4e$-3), Fgr pY511 (FC = 4.5; $P = 2.4e$-4), Protein Kinase C δ (Prkcd) pY311 (FC = 1.8; $P = 1.6e$-3), Ick pY159 (FC = 1.4; $P = 1.0e$-3), Mak pT157 (FC = 1.6; $P = 4.0e$-3), Pik3r1 pY467 (FC = 1.4; $P = 2.2e$-5), Inpp5d pY868 (FC = 2.4; $P = 6.7e$-9), and Siglec-F (Siglec5) pY561 (FC = 3.8; $P = 3.4e$-7). Among these proteins, Cdk1, Stat3, Brsk2, and Src have been previously reported to be activated during neurodegeneration (Counts & Mufson, 2017; Cavallini *et al*, 2013; Poulsen *et al*, 2017; Mota *et al*, 2014; Haim *et al*, 2015). Our data shed new light on their involvement at an early stage in this mouse model. Further, increased phosphorylation of specific Src-family kinases (Src, Lyn, Fgr) represents a novel insight into signaling pathway changes that have not been identified in other AD phosphoproteome datasets lacking pTyr enrichment. Overall, our phosphoproteomics analysis in CK-p25 mice shows that significant pTyr and pSer/pThr signaling changes are associated with neuroinflammation prior to neuronal loss during neurodegeneration.

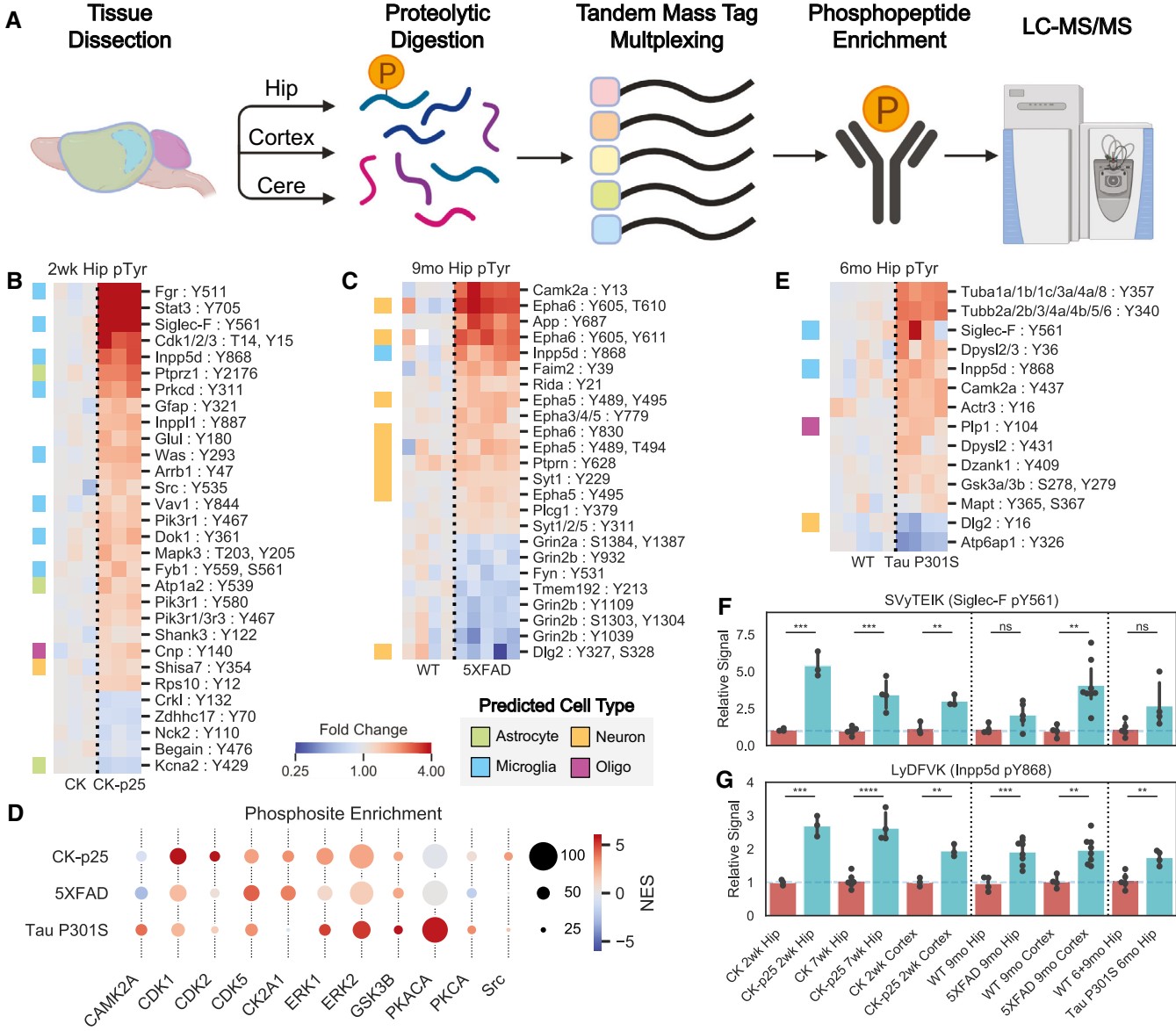

**Figure 1. Cross-model phosphoproteomics analysis identifies changes in signaling pathways.**

A    Workflow showing phosphopeptide enrichment strategy. Brain regions were dissected from mice and then processed into proteolytic peptide digests and labeled for quantification with TMT. Phosphopeptides were then enriched and analyzed by liquid chromatography followed by tandem mass spectrometry.

B, C    Heatmaps of enriched hippocampus phosphotyrosine peptides in hippocampus tissue from (B) CK-p25 and (C) 5XFAD animal mice. Colors indicate fold change relative to control animals on a log2-scale. Row colors (left) indicate peptides from predicted cell-type-specific proteins. Green = Astrocyte, Orange = Neuron, Blue = Microglia, Purple = Oligodendrocyte.

D    Phosphosite enrichment analysis of phosphopeptides from CK-p25, 5XFAD, and Tau mice. Circle colors indicate normalized enrichment scores (NES), and sizes indicate the number of observed sites in each substrate phospho-set within each mouse model. Kinases enriched in at least one model are shown.

E    Heatmap showing enriched hippocampus phosphotyrosine peptides from 6-month-old Tau P301S animals. Legend is same as (B).

F, G    Relative abundances across all tissues for phosphotyrosine peptides: (F) SVyTEIK (Siglec-F pY561); (G) LyDFVK (Inpp5d pY868). Bars indicate mean ± 95% confidence interval (CI); $n = 3$–8 samples; *$P < $5e-2, **$P < $1e-2, ***$P < $1e-3, ****$P < $1e-4; ns: not significant, using unpaired Student's *t*-test, two-sided.

## 5XFAD and Tau P301S phosphoproteomes tie unique signaling events to AD pathologies

We next assessed the 5XFAD mouse model of Aβ pathology (Oakley *et al*, 2006) to see if there were signaling changes that were shared in a second AD model. Using a the same phosphopeptide

enrichment methodology, we examined mice that were 9–10.5 months old with the aim of identifying signaling changes induced after pathology became pervasive in the hippocampus. This strategy led to the identification of 24,365 unique peptides (4% pTyr, 80% pSer/pThr) in the hippocampus and cortex of 5XFAD mice (5XFAD: $n = 8$; WT: $n = 4$; Dataset EV1). Of these peptides, 189 were

upregulated and 106 were downregulated in 5XFAD compared to WT mice (Fig EV1B and Fig EV2A). These changes included two peptides from the transgene amyloid precursor protein (APP): LVFFAEDVGSNK and HFEHVR (Fig EV2B). We identified pTyr sites that were upregulated on neuron-enriched proteins, including Arfgef3 pS1646 (FC = 1.7; $P$ = 6.9e-11), Epha5 pY489, pY495 (FC = 1.5; $P$ = 1.3e-9), Epha6 pY605, pT610 (FC = 2.6, $P$ = 3.0e-8), and Syt11 pS133 (FC = 1.7; $P$ = 7.8e-7), as well as microglia-enriched proteins Siglec-F pY561 (FC = 3.2; $P$ = 1.3e-3), Inpp5d pY868 (FC = 2.0; $P$ = 8.3e-10), Fyb1 pS561 (FC = 1.9; $P$ = 4.7e-5), Grp84 pS221 (FC = 1.6; $P$ = 6.0e-9), Selplg pS345 (FC = 1.6; $P$ = 2.9e-9), Dock2 pS1683 (FC = 2.2; $P$ = 3.9e-6), C5ar1 pS318 (FC = 2.3, $P$ = 7.8e-4), and Lsp1 pS243 (FC = 2.1; $P$ = 1.8e-7) (Fig 1C, and Fig EV2A and C). We observed some overlap in upregulated peptides between the hippocampus and cortex, although the correlation between tissue changes was lower than in CK-p25 (Fig EV1D and Fig EV2D). Motif analysis identified the enriched motif: -L-.-R-Q-.-pSer/pThr-[VLIMF]- within the set of downregulated peptides in the hippocampus (Fig EV2E). This motif closely matches the substrate motif of CaMKII, a kinase known to be dysregulated by Aβ in AD (White *et al*, 1998; Ghosh & Giese, 2015). Phosphosite enrichment analysis (PSEA) found that CaMKII-substrates were negatively associated with 5XFAD genotype (Fig 1D), further supporting decreased CaMKII activity in this model. We also observed that the downregulated phosphosites were enriched for proteins specifically expressed in neurons (Fig EV1E). Together, these data point to a model in which Aβ activates particular glial and neuronal signaling pathways and suppresses overall CaMKII activity in the hippocampus.

To determine the role of p25/Cdk5 signaling in 5XFAD AD pathogenesis, we compared the phosphoproteome of 5XFAD and WT mice to those crossed with mice harboring the Δp35KI mutation. Δp35KI prevents Cdk5 overactivation by blocking the generation of p25 from p35 by calpain in neurotoxic conditions (Lee *et al*, 2000; Seo *et al*, 2014). We found multiple Δp35KI-linked phosphorylation sites that were decreased in 5XFAD and corrected back to WT levels in 5XFAD;Δp35KI mice (Fig EV2F). Of these Δp35KI-linked sites, 62% matched the purported CaMKII phosphorylation motif in at least two adjacent residue positions compared to 16% pSer/pThr sites in background dataset. Intriguingly, we did not observe any effects of Δp35KI on markers of glial activation (Astrocyte: Aqp4, Gfap; Microglia: Ctsb, Ctsd, Siglec-F, Inpp5d) in 5XFAD mice. These findings suggest that Aβ activates microglia and astrocytes independent of p35 cleavage into p25, while suppressing signaling in neurons in a p25-dependent manner.

To assess signaling changes that occur in a Tau-centric model of AD and neurodegeneration, we quantified the phosphoproteome in Tau P301S mouse model (Ghetti *et al*, 2002). Using a similar phosphoproteomic platform, we collected pTyr, global pSer/pThr, and protein expression data from hippocampal tissues of 4-month-old mice when pathologies first develop. Interrogation of the 4 month Tau P301S dataset revealed minimal signaling from activated microglia, in agreement with disease progression in this model that suggests microgliosis occurs in grey matter at ~6 months of age (Yoshiyama *et al*, 2007; van Olst *et al*, 2020). We therefore also obtained hippocampal tissues from 6-month-old mice and quantified tyrosine phosphorylation to focus on signaling from activated microglia, as detected in our analysis of other models. In total, we identified 11,722 total unique peptides (6% pTyr, 83% pSer/pThr) across all tissues (P301S: 4 months $n$ = 5; 6 months $n$ = 4; WT: 4 months $n$ = 5, 6 months $n$ = 4, 9 months $n$ = 1; Dataset EV1). Of these peptides, 288 were upregulated while 32 were downregulated in Tau P301S mice compared to controls (Fig EV1B). We found that pTyr peptide abundance increased at both 4 and 6 months in P301S mice (Fig 1E and Fig EV2G). The global pSer/pThr phosphoproteome changes were dominated by increased phosphorylation of transgenic MAPT peptides (Fig EV2H and I) as well as members of the collapsin response mediator proteins (Crmp): Crmp1 pS8 (FC = 1.9; $P$ = 2.0e-7), Crmp2 (Dpsyl2) pY499, pT509 (FC = 2.0; $P$ = 2.0e-5), Crmp4 (Dpsyl3) pY499, pT509 (FC = 1.9; $P$ = 2.9e-7), and Crmp5 (Dpsyl5) pS536 (FC = 2.4; $P$ = 1.7e-5). We also observed increased phosphorylation on cytoskeletal proteins Tubulin alpha (Tuba1a/1b/1c/3a/4a/8) pY357 (FC = 2.0; $P$ = 6.1e-8), Tubulin beta (Tubb2a/2b/3/4a/4b/5/6) pY340 (FC = 2.0; $P$ = 1.1e-7), Nefm pS550 (FC = 1.6; $P$ = 3.3e-3), and Map1b pT1312, pT1336 (FC = 1.4; $P$ = 1.4e-3), as well as synapse-associated proteins Dlg2 pY708 (FC = 1.6; $P$ = 8.2e-4), Dlg4 pY580 (FC = 1.5, $P$ = 1.4e-3), Tanc2 pS140 (FC = 1.6; $P$ = 8.8e-5), and Bcan pS546 (FC = 1.4; $P$ = 7.4e-3). Peptides that were upregulated at 4 months were similarly upregulated at 6 months (Fig EV2J). However, we saw upregulation of two microglia-enriched pTyr peptides, Siglec-F pY561 (FC = 2.1; $P$ = 0.10) and Inpp5d pY868 (FC = 1.6; $P$ = 6.4e-3) at 6 months (Fig 1E). In contrast with the other models, we did not identify any enriched phosphorylation motifs in P301S mice. Together, our phosphoproteome datasets allow us to compare and contrast signal networks in different neurodegenerative processes.

## Cross-model comparison identifies shared kinase and microglial activation

To identify central signaling nodes associated with AD and neurodegeneration, we searched for common changes across mouse models. Phosphosite enrichment analysis (PSEA) identified signatures of shared kinase activation for Cdk1/2/5, extracellular signal-regulated kinase 1/2 (Erk1/2), and Gsk3β (Fig 1D). Among these kinase family members, we observed significant upregulation of Cdk5 protein (FC = 1.3; $P$ = 1.9e-4) as well as phosphosite upregulation on Erk1 pT203, pY205 (FC = 1.4, $P$ = 1.8e-3), Cdk1/2/3 pT14 (FC = 6.2, $P$ = 2.7e-3) and pY15 (FC = 3.7, $P$ = 3.1e-3), and Src pY535 (FC = 1.5, $P$ = 5.4e-3) in 2wk CK-p25 mice. We additionally observed increased phosphorylation on Gsk3 (Gsk3a pS278, pY279/ Gsk3b pS215, pY216; FC = 1.3, $P$ = 2.7e-3) in Tau P301S mice at 4 months (Dataset EV1). Comparing cell-type categories across all models, we find that CK-p25 and 5XFAD have significant microglial-enriched phosphoproteome changes (Fig EV1E). To further define the signaling pathways that are disrupted in each model, we applied gene ontology enrichment analysis to the phosphoproteomes of each mouse model (Dataset EV2). The upregulated pathways in CK-p25 mice included cytoskeleton organization and cell cycle/DNA replication signaling (Fig EV3A). In 5XFAD mice, the upregulated pathways included endosome and synaptic vesicles, while the downregulated pathways included ion transporter and receptor complexes (Fig EV3B). We only found upregulated pathways enriched in the Tau P301S mice, primarily involving the cytoskeleton and microtubules (Fig EV3C). In total, our systems analysis finds that the diseased mouse models have numerous kinases,

cell-type-specific, and cytoskeletal signaling pathways disrupted at the level of the phosphoproteome.

We next compared individual peptides that were identified across all AD models to find peptides associated with neurodegenerative processes. Out of 30,370 total unique peptides, 3,684 peptides (8% pTyr, 88% pSer/pThr) were identified in all three animal models. We observed shared upregulation of phosphorylation on Stat3, Prdx1, Rida, Gpr84, and Trim3 in CK-p25 and 5XFAD mice (Fig EV1B). We additionally found overlapping phosphorylation increases on Crmp1, Dpysl5, Pafah1b2, and Ank2 in CK-p25 and

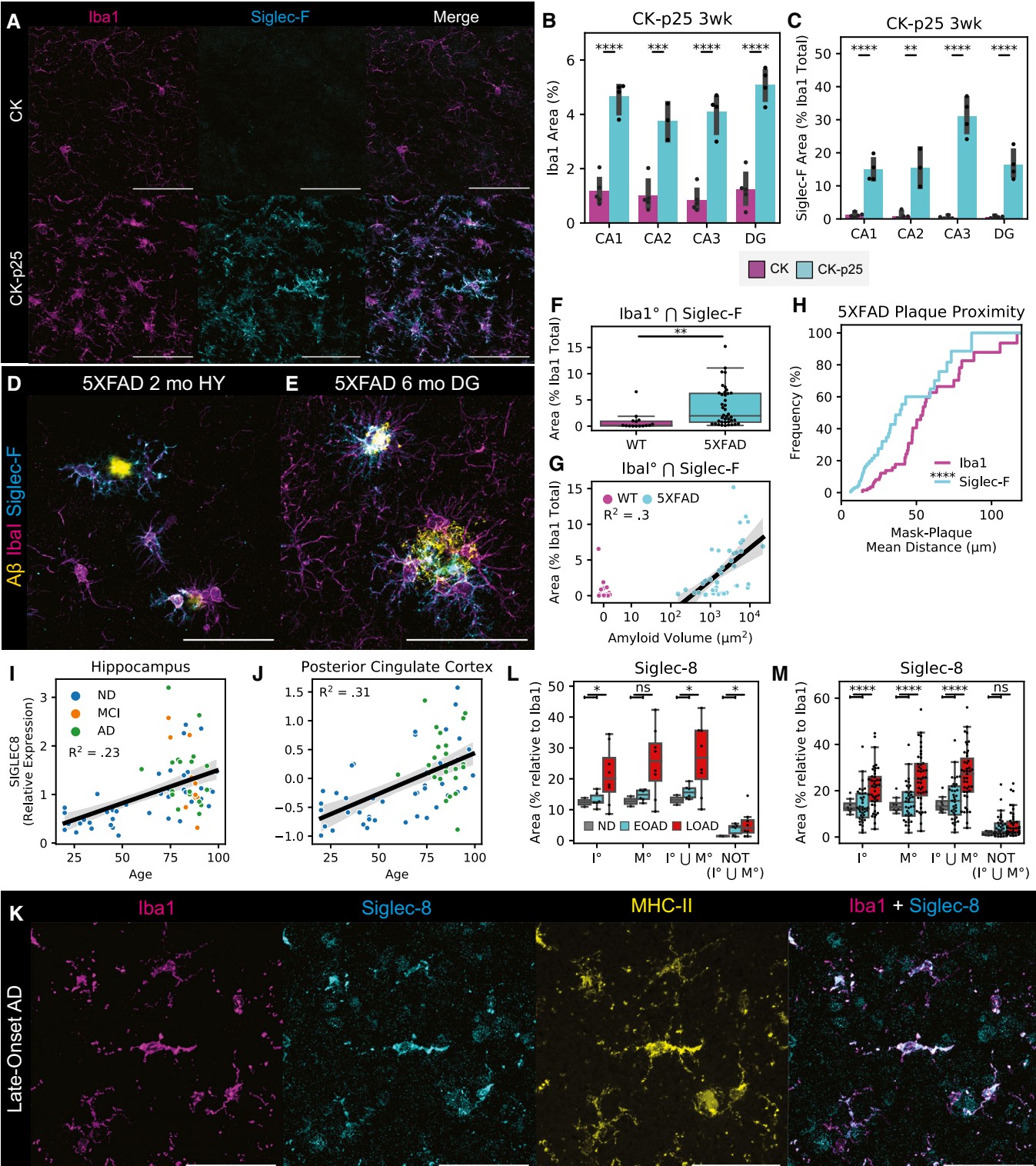

Figure 2.

Figure 2.  Siglec-F and Siglec-8 are upregulated in microglia across models of neurodegeneration.

A       Immunofluorescence (IF) staining showing Siglec-F and Iba1 localization in hippocampus CA3 region of 3wk induced CK (top) or CK-p25 (bottom) mice. Scale
        bars = 50 μm. Colors are: Magenta = Iba1, Cyan = Siglec-F. Images are 60× super-resolution max z-stack projections taken from coronal slices.
B, C    Percent area coverage of (B) Iba1 and (C) Iba1-proximal Siglec-F between CK and CK-p25 mice across hippocampal regions. Bars indicate mean ± 95% CI; n = 3–4
        animals; *$P < 5e-2$, **$P < 1e-3$, ***$P < 1e-3$, ****$P < 1e-4$; ns: not significant, using unpaired Student's $t$-test, two-sided.
D, E    IF staining of Siglec-F and Iba1 in (D) 5XFAD 2 month hypothalamus (HY) and (E) 5XFAD 6 month dentate gyrus (DG). Yellow = Aβ from methoxy X04 (D) or D54D2
        (E). Scale bars = 50 μm.
F       Percent area coverage of Iba1-proximal Siglec-F relative to total Iba1 across all analyzed thalamic, cortical, and hippocampal regions of 6-month WT (non-Tg) and
        5XFAD mice. Box indicates quartiles, and whiskers indicate the last datum within 1.5 inter-quartile range; n = 15–40 images (3–5 animals); *$P < 5e-2$, **$P < 1e-2$,
        ***$P < 1e-3$, ****$P < 1e-4$; ns: not significant, using unpaired Student's $t$-test, two-sided.
G       Percent area coverage of Siglec-F localized to Iba1 compared to total Aβ plaque volume in each field of view across all analyzed images from 5XFAD and non-Tg
        control mice. Linear regression $R^2$ value is shown for images with ≥ 10 μm$^2$ total amyloid volume.
H       Cumulative distribution function (CDF) of the distance between Siglec-F$^+$ and Iba1$^+$ mask voxels and nearest Aβ plaques in 6-month 5XFAD mice. Each value in the
        CDF represents the mean distance calculated from a single z-stack of a ROI containing ≥ 1 plaque of size ≥ 10 μm$^2$. n = 55 images (4 animals).
I, J    Relative RNA abundances for Siglec-8 in (H) hippocampus and (I) posterior cingulate cortex postmortem patient tissue. Linear regression $R^2$ values are shown on
        plots.
K       IF staining showing Siglec-8 and Iba1, and MHC-II in cortical white matter tissue from a patient with late-onset AD (20× wide-field; Case ID: 01-43). Scale
        bars = 50 μm. Images are 60× super-resolution max z-stack projections.
L, M    Percent area coverage of Siglec-8 localized to Iba1 or MHC in individuals with non-AD or AD. In (K), each point represents one WM or GM region average from each
        individual. In (L), values and statistics are shown for each image analyzed. Boolean mask operators indicate: ° = mask dilation, ∩ = intersection, ∪ = union,
        NOT = inverse mask. Box indicates quartiles, and whiskers indicate the last datum within 1.5 inter-quartile range; n = 8–43 images (n = 1 ND, 3 early-onset AD, 4
        late-onset AD); *$P < 5e-2$, **$P < 1e-2$, ***$P < 1e-3$, ****$P < 1e-4$; ns: not significant, using unpaired Student's $t$-test, two-sided.

Tau P301S mice. Between 5XFAD and Tau P301S mice, only Bcan had shared phosphorylation increases. Finally, we identified two phosphopeptides that were consistently upregulated in diseased models: Siglec-F pY561 and Inpp5d pY868 (Fig 1F and G). We manually validated these spectra to confirm the peptide assignments were correct (Fig EV3D and E). Both of these proteins are expressed primarily in microglia and are related to human genes *CD33* and *INPP5D* identified in AD GWAS studies (Karch & Goate, 2014). Due to the underexplored nature of Siglec-F in neurodegeneration, we next investigated its role in microglial activation.

## Siglec-F is upregulated on subsets of inflammatory microglia

To determine whether increased Siglec-F phosphorylation was associated with increased Siglec-F protein expression in our models, we used immunofluorescence (IF) imaging to locate and quantify Siglec-F expression. We first validated that the antibody E50-2440 was able to label Siglec-F in BV-2 cells with stable viral protein expression of Siglec-F carrying mutated pTyr sites, with minimal immunoreactivity in control cells expressing an empty vector (Fig EV4A). We next stained CK and CK-p25 brain slices from mice after 3 weeks of p25 induction. We observed that a subset of Iba1$^+$ microglia in CK-p25 mice stained positive for Siglec-F expression (Fig 2A). In contrast, we did not observe immunoreactivity for Siglec-F on microglia in CK control mice that lack p25 expression. As previously observed, CK-p25 microglia exhibited gliosis (Mathys *et al*, 2017), quantified here as an increase in the area covered by Iba1 signal (Fig 2B). Relative to these changes, Siglec-F was upregulated throughout the dentate gyrus (DG), Cornu Ammonis 1 (CA1), CA2, and CA3 regions of the hippocampus in CK-p25 mice (Fig 2C, and Fig EV4B and C). Across all four regions, ~15–35% of area was positive for Siglec-F relative to Iba1 and this amount was highest in the CA3. We quantified the % area overlap of Siglec-F and Iba1 in CK-p25 mice and found that 97% of Siglec-F signal was located ≤ 2 voxels away from Iba1 (Fig EV4D and E). Together, our findings show that Siglec-F is expressed on a subset of activated microglia in the CK-p25 mouse model.

We next stained for Siglec-F and Iba1 expression in 5XFAD mouse brain tissue. To understand if these microglia are linked to early Aβ deposition, we stained the hypothalamus of 2-month-old 5XFAD mice where Aβ plaques are first observed (Canter *et al*, 2019). We observed microglia in this region that expressed Siglec-F in proximity to Aβ plaques (Fig 2D). By 6 months, we found Siglec-F$^+$ microglia throughout the thalamus, hippocampus, and cortex of 5XFAD mice (Fig 2E and F, and Fig EV4F). The amount of Siglec-F was positively correlated with the total volume of Aβ plaques in each field of view (Fig 2G). As in CK-p25 mice, > 99% of all Siglec-F signal was located ≤ 2 voxels away from Iba1 (Fig EV4G and H). Similar to CK controls, we observed minimal Siglec-F signal in 6-month-old WT mice (Fig 2F). To test whether these Siglec-F$^+$ microglia were plaque-associated, we calculated the mean distance between Siglec-F and Iba1 mask voxels and neighboring Aβ plaques in wide-field z-stack images from plaque-containing regions of 6-month 5XFAD animals. We observed that Siglec-F$^+$ voxels were significantly closer to Aβ plaques compared to total Iba1 (Fig 2H). Together, these data show that Siglec-F is upregulated on microglia in response to Aβ pathology and may play a role in microglia activation.

To further explore the potential relationship between Siglec-F and microglial activation, we analyzed published single-cell and bulk RNA-seq datasets. In CK-p25 mice (Mathys *et al*, 2017), we observed three Siglecs (*Siglec-1*, *Cd22*, and *Siglec-F*), upregulated in late-response reactive microglia that appear by two weeks of p25 induction (Fig EV5A). Other Siglecs (*Cd33*, *Siglec-E*, *Siglec-G*, *Siglec-H*) were downregulated in late-response microglia when compared with homeostatic microglia from CK controls. A subset of these reactive microglia that express type-II interferon response genes (marker genes: *Lpl*, *Spp1*) showed highest Siglec-F expression. We next examined the disease-associated microglia (DAM; marker genes: *Lpl*, *Spp1*) population identified in 5XFAD mice (Keren-Shaul *et al*, 2017) and observed a similar trend: Cd22 and Siglec-F were upregulated in DAM compared to homeostatic microglia (Fig EV5B). Moreover, a gene co-expression network analysis found that Siglec-F expression increased in the entorhinal cortex of 4- to 6-month-old

Tau P301L (rTg4510) mice (Castanho *et al*, 2020). Finally, Siglec-F was found to be upregulated in the hippocampus of mice at 24–-29 months compared to 3 months (Stilling *et al*, 2014). These results suggest that activated microglia express Siglec-F in brain regions associated with aging and neurodegeneration.

### Siglec-8 is upregulated in aging-associated microglia

To identify the human receptor analog of mouse Siglec-F, we examined published single-cell sequencing datasets. While CD33 has been previously identified in IHC staining of AD patient tissue (Griciuc *et al*, 2013), other related receptors Siglec-5 and Siglec-8 have not been carefully examined. Siglec-5 shares the closest sequence homology with Siglec-F (Angata *et al*, 2001; Aizawa *et al*, 2003), however, Siglec-8 more closely matches the expression and substrate binding patterns of Siglec-F on eosinophils (O'Sullivan *et al*, 2018). A single-cell analysis on temporal lobe biopsies identified two clusters of major histocompatibility complex II+ (MHC-II+) aging-associated microglia (Sankowski *et al*, 2019). Siglec-8 was classified as a marker gene for one of these aging-associated microglia populations and was upregulated only in these two clusters. An additional single-nuclei RNA-seq analysis of human AD (Zhou *et al*, 2020) found an AD-associated microglia cluster that had increased expression of Siglec-8. We examined bulk tissue RNA microarray data from young and aging brain tissues obtained from a previously published dataset (Berchtold *et al*, 2008). We found that Siglec-8 had increased expression in individuals over the age of 60 in the hippocampus and other brain regions (Fig 2I and J, and Fig EV5C and D). Among individuals with age ≥ 60 we observed a trend toward increased Siglec-8 transcript abundance in AD, but this effect was not significantly different from MCI and ND cases. These data suggest that Siglec-8 expression may be upregulated in aged human microglia, while little is known about Siglec-5 due to its low expression levels.

To explore whether Siglec-8+ microglia are present in AD, we used IF to quantify Siglec-8 expression in human postmortem cortical tissues. We first stained for Siglec-5 and Siglec-8 and observed that Siglec-8 signal was co-localized with Iba1 (Fig EV6A). We showed that this Siglec-8 antibody (ab198690) was able to label BV-2 cells expressing Siglec-8 (Fig EV6B). We then stained for Siglec-8, MHC-II, and Iba1 in human cortical sections containing both white matter (WM) and grey matter (GM) regions. We analyzed tissue from a healthy control, early-onset AD patients and late-onset AD patients ($n = 1$, 58 years, Braak I; $n = 3$, 60 years, Braak VI; $n = 4$, 77–96 years, Braak IV-V; Dataset EV3). We observed Siglec-8 signal co-localized with Iba1+/MHC-II+ microglia in early- and late-onset AD patients (Fig 2K and Fig EV6C–E). For each image, we generated fluorescent signal masks for quantification of channel overlap. We observed that 75% of Siglec-8 signal was located ≤ 2 voxels away from Iba1 or MHC-II in images with minimal blood vessel autofluorescence (Fig EV6F and G). We then compared percent area of Siglec-8 localized to Iba1 across patient groups and found increased overall Siglec-8 expression in late-onset AD compared with early-onset and control individuals (Fig 2L and M). This comparison was significant both when comparing individual image stack averages as well as patient averages from GM and WM regions. Although Siglec-8 expression was increased in some brain regions of early-onset AD patients, this effect was not significant compared to the control

patient samples. Siglec-8+ microglia did not appear to be significantly biased toward either WM or GM regions (Fig EV6H). The amount of Siglec-8 also did not appear to be correlated with the total volume of Aβ plaques in a field of view (Fig EV6I). Altogether, these results show that human microglia upregulate Siglec-8 in an age-dependent manner.

### Siglec-F and Siglec-8 can be upregulated by IFNγ

Given that Siglec-F+ microglia were enriched for type II interferon response genes in the CK-p25 model, we next sought to understand whether Siglec-F and Siglec-8 are regulated by inflammatory cytokine signaling. Previous work has identified the cytokine interferon gamma (IFNγ) as a regulator of type II interferon inflammation in AD patients and mouse models of neurodegeneration (Mastrangelo *et al*, 2009; Wood *et al*, 2015; Zheng *et al*, 2016) and a potential driver of Siglec-8 expression in eosinophils (Steinke *et al*, 2013). Using flow cytometry, we found that BV-2 cells treated with 10 ng/ml IFNγ for 72 h had increased Siglec-F expression (Fig 3A). In this system, we observed that cells had increased size, reduced cell count, and reduced doubling times after treatment with IFNγ (Fig EV7A–C). The effects on cell size and proliferation were observed in treatments between 1 and 100 ng/ml IFNγ, suggesting that we are saturating IFNγ receptor signal. IFNγ is known to signal through Janus kinases (Jak1 and Jak2), which can be targeted with the small molecule inhibitor tofacitinib (Flanagan *et al*, 2010). The effects of IFNγ on cellular phenotypes and Siglec-F expression were blocked by co-treating cells with ≥ 5 μM tofacitinib (Figs 3A and EV7D and E). Together, these data show that Siglec-F expression can be upregulated by IFNγ and mediated through the Jak1/2 pathway.

We next tested the effect of IFNγ on an *in vitro* human induced pluripotent stem cell-derived microglia (iMGL) model. We observed that iMGLs upregulated Siglec-8 expression after 72 h of treatment with 25 ng/ml IFNγ by IF (Fig 3B). This effect was less effective with IFNα or IFNβ, suggesting type-II interferon signaling dependence (Fig 3C). We observed that co-treatment of iMGLs with IFNγ and 5 μM tofacitinib was able to reverse the upregulation of Siglec-8 by IF and qPCR (Figs 3D and EV7F and G). We were able to replicate these effects in iMGLs matured in the presence of homeostasis-inducer transforming growth factor beta (TGFβ; Butovsky *et al*, 2014; Fig EV7H). In summary, our results support the link between Siglec-F and Siglec-8 expression with type II interferon-activation states.

### Siglec expression in BV-2 cells induces cell death

To explore the functional relevance of expression and phosphorylation of Siglec-F and related receptors, we designed a Siglec-overexpression system using BV-2 cells. Using a MSCV-IRES-puro (MIP) retrovirus we found that constitutive overexpression of Siglec-F increases culture doubling times (Fig 4A). This effect was dependent on Siglec-F's cytosolic tyrosine residues, as individual Y561F or Y538F mutations led to a partial loss-of-function (LOF), while the Y538F;Y561F (2xY->F) mutant showed complete LOF (Fig 4A). Increased doubling times were associated with ligand binding, as co-treatment of cells with 50 mU sialidase was able to partially restore proliferation in WT Siglec-F cells. However, sialidase treatment was slightly toxic to cells carrying LOF mutations as

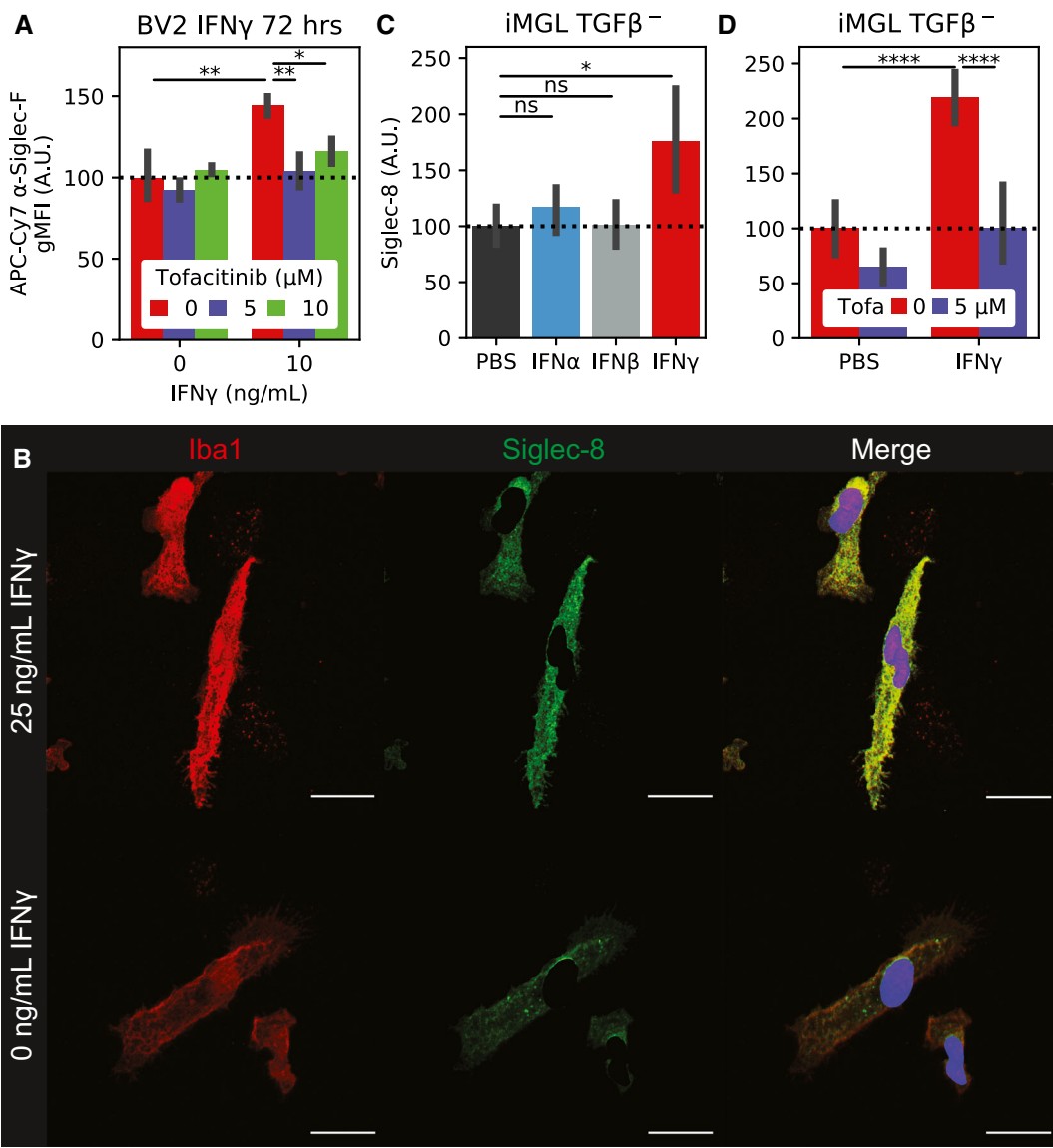

**Figure 3. Siglec-F and Siglec-8 are upregulated in BV-2 cells and iMGLs by IFNγ stimulation.**

A Flow cytometry quantification of BV-2 Siglec-F expression levels. Cells were co-treated with 0-10 ng/ml IFNγ and 0-10 μM tofacitinib. Values are geometric mean fluorescence intensities (gMFI). Bars indicate mean ± 95% CI; n = 2-4 replicates; *P < 5e-2, **P < 1e-2, ***P < 1e-3, ****P < 1e-4; ns: not significant, using unpaired Student's t-test, two-sided. Dotted line indicates mean of untreated group.

B IF staining of iMGLs treated with IFNγ. Siglec-8 fluorescence is only shown on non-nuclear (Hoechst 33342⁻) regions. Images are 60× super-resolution max z-stack projections. Colors are as follows: Red = Iba1, Green = Siglec-8, Blue = 33342. Scale bars = 20 μm.

C Quantification of Siglec-8 on iMGLs treated with PBS or 25 ng/ml IFNα, IFNβ, or IFNγ. Values are mean Siglec-8 intensity values from Iba1⁺;33342⁻ regions. Bars indicate mean ± 95% CI; n = 8 images (2 replicates); *P < 5e-2, **P < 1e-2, ***P < 1e-3, ****P < 1e-4; ns: not significant, using unpaired Student's t-test, two-sided. Dotted line indicates mean of untreated group.

D Quantification of Siglec-8 on iMGLs treated ± 25 ng/ml IFNγ and 5 μM tofacitinib. Legend is same as (C). Bars indicate mean ± 95% CI; n = 5-20 images (2 replicates); *P < 5e-2, **P < 1e-2, ***P < 1e-3, ****P < 1e-4; ns: not significant, using unpaired Student's t-test, two-sided. Dotted line indicates mean of untreated group.

measured by an increase in doubling time (Fig 4A). Previous studies suggest that Siglec activation can trigger apoptotic events in eosinophils (Zhang *et al*, 2007; Zimmermann *et al*, 2008; Kiwamoto *et al*, 2015; Youngblood *et al*, 2019). Using flow cytometry, we observed an increased proportion of BV-2 cells expressing WT or Y561F Siglec-F that were labeled with early (Annexin-V⁺) and late

apoptotic (Annexin-V+; PI⁺) markers (Fig 4B). Thus it appears that constitutive overexpression of Siglec-F activates a cellular death program such as apoptosis, necroptosis, pyroptosis, or ferroptosis (Green, 2019).

We repeated these experiments using the human Siglec receptors: (i) CD33, which is implicated in AD GWAS studies (Karch &

Goate, 2014); (ii) Siglec-5, which has the closest sequence homology to Siglec-F; (iii) Siglec-8, which has the closest substrate binding pattern to Siglec-F (preferring 6'-sulfo sialyl Lewis x; O'Sullivan *et al*, 2018). Overexpression of each human Siglec caused a similar increase in doubling times, while 2xY->F mutants (CD33 Y340F; Y358F, Siglec-5 Y520F;Y544F, and Siglec-8 Y447F;Y470F) did not affect growth rates (Figu 4A). Co-treatment with 50 mU sialidase was able to restore proliferation in CD33-expressing cells to a similar degree as Siglec-F. However, the effect of sialidase was not as strong in cells expressing Siglec-5 or Siglec-8, suggesting a possible

difference in substrate binding patterns necessary for Siglec activation (Fig 4A). These results indicate that Siglec-F and related human Siglec receptors may signal in a similar ITIM-dependent manner to induce cell death, although their activating context may vary slightly.

To further validate these findings, we repeated these experiments in BV-2 cells using the pINDUCER dox-inducible lentiviral toolkit (Meerbrey *et al*, 2011). Expression of Siglec-F in this system showed dose-dependent impairment of proliferation with differences in cell count and confluency within 24 h after addition of 500 ng/ml

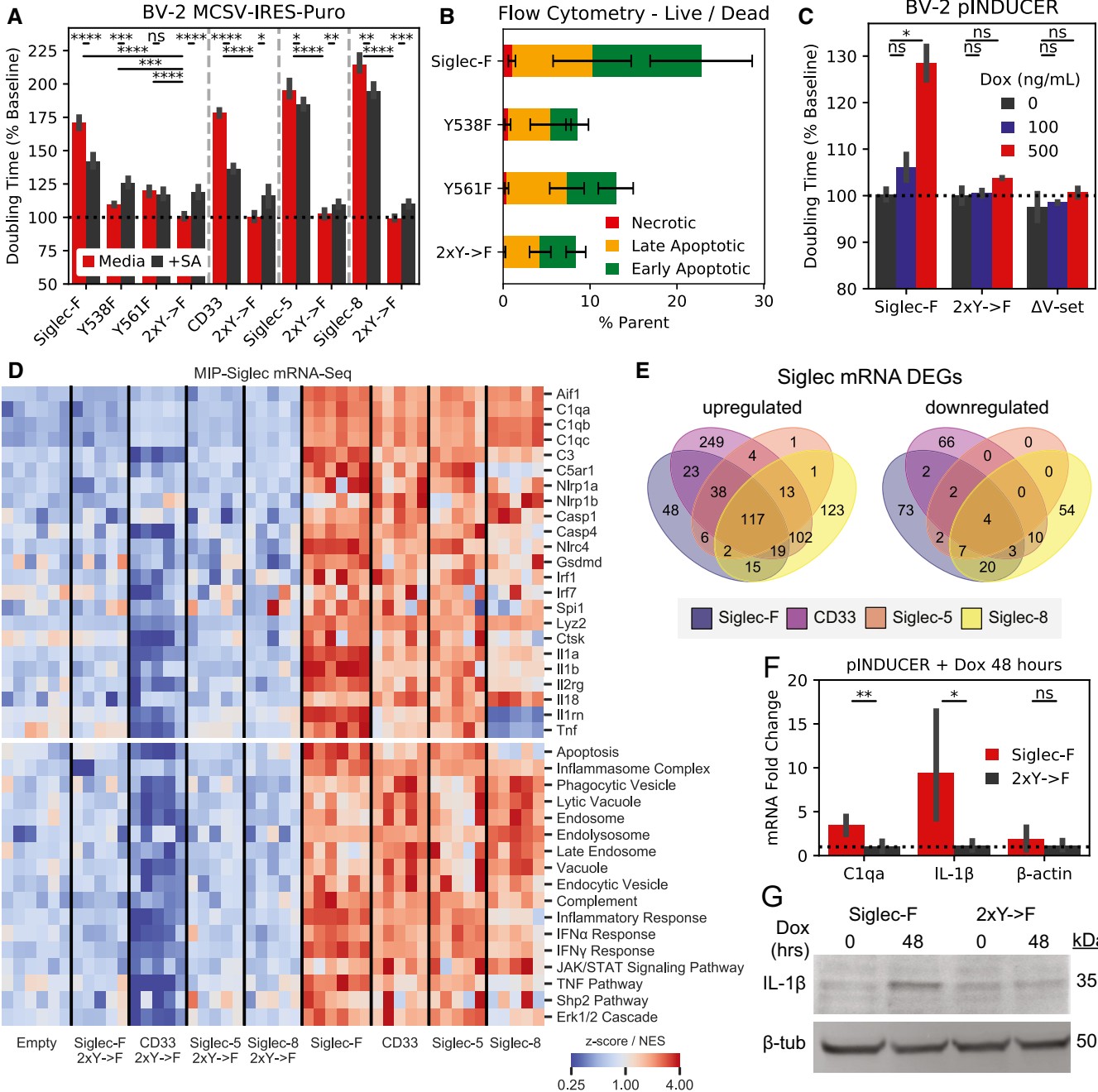

**Figure 4.**

**Figure 4. Siglec-F and related human Siglecs drive a pro-inflammatory response in BV-2 cells.**

A   Confluency doubling times quantified from BV-2 cells with stable retroviral expression of Siglec-F and related human Siglec constructs. Plots indicating doubling time (normalized to 2xY->F constructs) estimated from Incucyte brightfield images. Cells grown in media alone or + 50 mU sialidase (+SA). Bars indicate mean ± 95% CI; n = 3-12 replicates; *$P$ < 5e-2, **$P$ < 1e-2, ***$P$ < 1e-3, ****$P$ < 1e-4; ns: not significant, using unpaired Student's $t$-test, two-sided. Dotted line indicates mean of 2xY->F groups.

B   Flow cytometry quantification of cell live/ dead markers Annexin-V and propidium iodide on BV-2 cells with stable retrovirus expression of Siglec-F constructs. % parent values are shown for early apoptotic (Annexin-V$^+$;PI$^-$), late apoptotic (Annexin-V$^+$;PI$^-$), and necrotic (Annexin-V$^-$;PI$^+$) populations. Bars indicate mean ± SD; n = 8 replicates.

C   Confluency doubling times quantified from BV-2 cells with dox-inducible expression of Siglec-F constructs. Plots indicating doubling time (normalized to 2xY->F constructs) estimated from Incucyte brightfield images. Bars indicate mean ± SD; n = 8 replicates; *$P$ < 5e-2, **$P$ < 1e-2, ***$P$ < 1e-3, ****$P$ < 1e-4; ns: not significant, using unpaired Student's $t$-test, two-sided. Dotted line indicates mean of 2xY->F group.

D   Selected gene expression changes (top) and enriched gene sets (bottom) taken from RNA-seq analysis of BV-2 cells with stable retroviral expression of Siglec constructs, 2xY->F, and empty vector controls. Heatmaps share a color map range of 0.25 to 4 for z-score values (top), normalized enrichment scores (NES, bottom).

E   Quantification and overlap of differentially expressed genes (DEGs) between each Siglec-expressing BV-2 cell line. DEGs were calculated for each Siglec gene compared to respective 2xY->F mutants (upregulated $P$ < 1e-8, FC> 1.5; downregulated $P$ < 1e-8, FC < 0.66).

F   Expression levels of C1qa, IL-1β, and β-actin by qPCR analysis in BV-2 cells with inducible Siglec-F expression after 48 h of treatment with 500 ng/ml doxycycline. Fold Change values are equal to $2^{\Delta\Delta Cq}$ normalized relative to GADPH mRNA levels. Bars indicate mean ± 95% CI; n = 6 replicates; *$P$ < 5e-2, **$P$ < 1e-2, ***$P$ < 1e-3, ****$P$ < 1e-4; ns: not significant, using unpaired Student's $t$-test, two-sided. Dotted line indicates mean of 2xY->F group.

G   Western blot showing pro-IL-1β and β-tubulin in BV-2 cells induced to express Siglec-F at 0 and 48 h.

doxycycline (Figs 4C ad EV8A). Previous literature suggests that that the N-terminal V-set domain of Siglecs is necessary for sialic acid substrate binding (Griciuc *et al*, 2013). We tested an additional construct of Siglec-F in this system that has its V-set binding domain deleted (Siglec-F ΔV-set: ΔD18-D116). Siglec-F ΔV-set had the same growth rate as 2xY->F cells (Fig 4C), suggesting a similar LOF caused by impaired substrate binding. Together our findings indicate that Siglec-F ligand binding and ITIM phosphorylation are necessary to induce cell death.

**ITIM-containing Siglecs activate inflammatory signaling pathways**

To gain further insight into the mechanistic effects of Siglec activation, we examined gene expression changes in BV-2 cells overexpressing Siglec-F or related human Siglecs using bulk RNA-seq. Downstream analysis identified overlapping transcriptional network alterations induced by all four Siglec receptors (Fig 4D, Dataset EV4), including components of the complement system (*C1qa, C1qb, C1qc, C3*), components of the inflammasome (*Nlrp1a, Nlrp1b, Casp1, Casp4, Gsdmd*), pro-inflammatory cytokines (*Il1a, Il1b, Il18, Tnf*), and lysosomal proteins (*Lyz1, Lyz2*) (Fig 4D). This could potentially explain the mechanism of Siglec-induced cellular death, as the inflammasome is known to regulate pyroptosis (Green, 2019). In contrast, 2x->F mutants and empty vector controls did not induce transcriptional changes for these genes, suggesting a critical role for ITIM phosphorylation in driving gene expression changes. Close to half of all Siglec-F upregulated genes were also induced by all 3 of the human Siglecs (Fig 4E). Siglec-F and Siglec-5 shared the most similar gene expression changes (90% differentially expressed gene (DEG) overlap), while CD33 and Siglec-8 both induced a unique set of DEGs (32% and 38% shared DEGs with Siglec-F, respectively). We next sought to validate whether the shared signaling pathways involving the inflammasome could be reproducibly activated by Siglec-F. We validated that Siglec-F drove increased IL-1β production after dox induction using qPCR (Fig 4F) and western blot (Fig 4G) analyses. Previous literature has found that aminopeptidase inhibitor bestatin can block anthrax lethal toxin-induced cell

death via NLRP1 (Wickliffe *et al*, 2008; Chui *et al*, 2019). We found that bestatin was able to partially reverse the effects of Siglec-F on BV-2 proliferation in our growth curve assay (Fig EV8B and C). In summary, our results show that expression of Siglec-F and related human Siglecs activates an inflammatory signaling response in BV-2 cells.

To understand the signaling networks connecting Siglec ITIM phosphorylation with downstream gene expression changes, we applied pTyr analysis to BV-2 cells expressing Siglec-F and related human Siglecs (Fig 5A, Dataset EV5). We found that Siglec-F drove reduced phosphorylation on a large number of proteins (Fig 5B), consistent with its role as an inhibitory receptor. These downregulated phosphopeptides included proteins associated with cellular adhesion, metabolic processes, signal transduction, cellular cytoskeleton, and endosomal trafficking. However, we also observed several pTyr sites on proteins involved in signal transduction that were increased in abundance after Siglec-F expression (Fig 5C). Among these sites, we observed that Inpp5d pY868, Prkcd pY311, Inppl1 pY887, Dok1 pY361, Lyn pY508, Fyb1 pY559 pS561, Erk1 (Mapk3) pT203 pY205, and Pik3r1 pY508 has increased phosphorylation in our CK-p25 model (Fig 5D). This signaling response appeared to be due to ITIM phosphorylation and was shared among human Siglecs, as several of these phosphopeptides had a common directional change in response to CD33, Siglec-5, and Siglec-8 expression (Fig 5E and F). To validate that these signaling changes do not occur during generation of stable cell line, we selected Erk1/2 (Mapk3/1) as targets and blotted for phospho-Erk activation in BV-2 cells with inducible Siglec-F expression. We observed that Erk1 and Erk2 had increased phosphorylation after 48 h of dox induction of Siglec-F but not 2xY->F mutants (Fig EV8D and E). Myc-tagged Siglec-F itself was expressed as a 70 kDa glycoprotein that was downregulated after 48 h of induction in response to its own activation (Fig EV8D and F). In support of our transcriptomic data, we also saw an overlapping set of proteins that increased in the global proteome analysis of BV-2 cells (Fig EV8G). In addition, we observed the Siglec-F pY561 phosphopeptide (SVyTEIK) was specifically increased in Siglec-F expressing cells (Fig EV8H). Together, our transcriptomic, proteomic, and pTyr analyses suggest that Siglec-F may play a dual

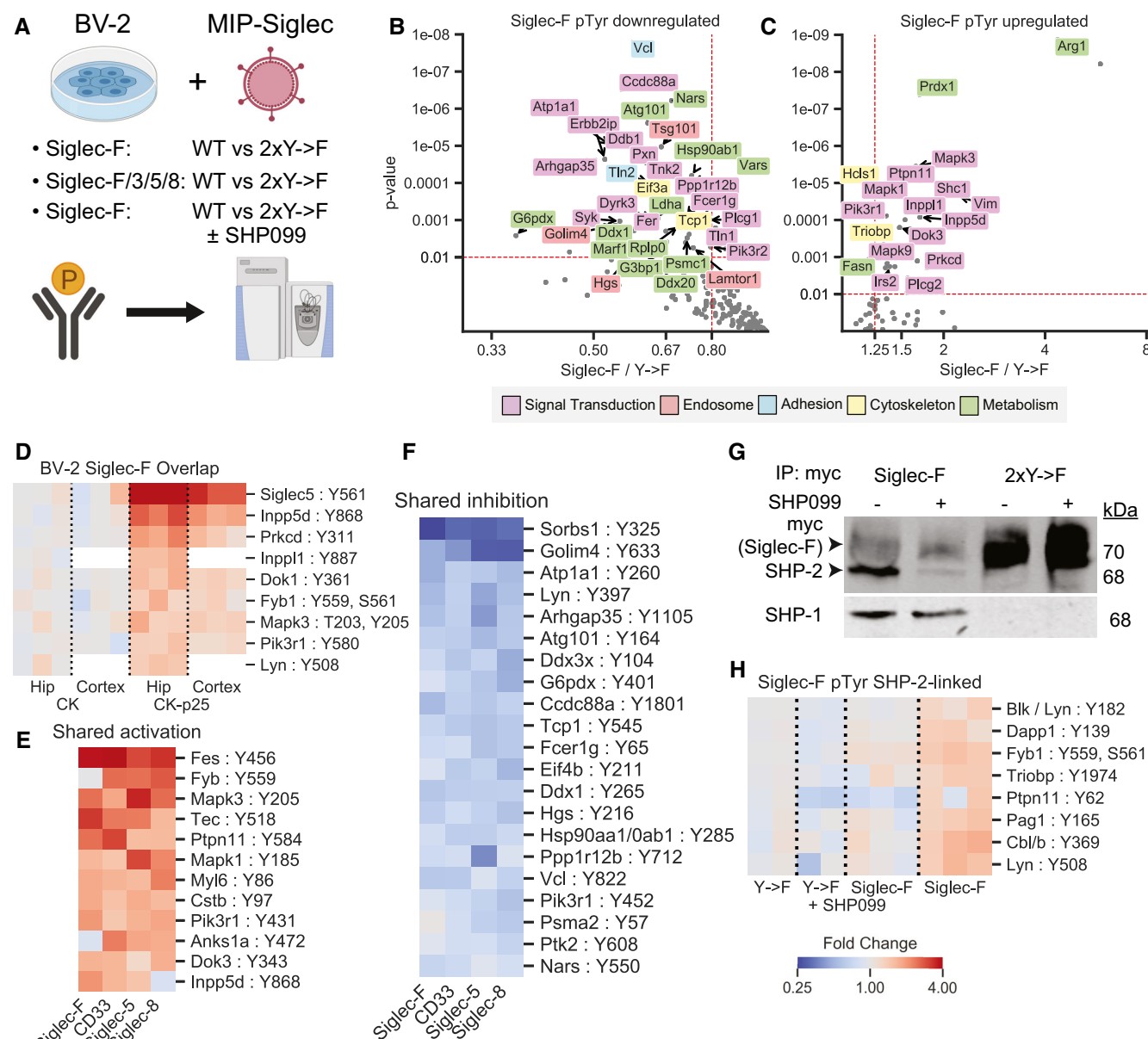

**Figure 5. Siglec-F perturbs phosphotyrosine signaling pathways in BV-2 cells.**

A   Experimental design for phosphotyrosine analysis of BV-2 cells with stable expression of Siglec and corresponding 2xY->F mutants. Samples were run as three 10-plex experiments: (1) Siglec-F: WT vs. 2xY->F; (2) Siglec-F, CD33, Siglec-5, and Siglec-8: WT vs 2xY->F; (3) Siglec-F: WT vs. 2xY->F ± 1 µM SHP099. Phosphotyrosine peptides were enriched and analyzed by LC-MS/MS.

B, C   Volcano plots showing phosphotyrosine peptides that were (B) increased or (C) decreased after Siglec-F expression. Plots show Siglec-F compared to 2xY->F mutant data integrated across all three runs. Protein names are shown for altered peptides. Labels are only shown for peptides with maximum directional change from each protein. Labels colored according to GO terms: magenta = signal transduction, red = endosome, cyan = cell adhesion, yellow = cytoskeleton, green = metabolism.

D   Heatmap showing phosphosites in CK-p25 which overlapped in their directional change with BV-2 Siglec-F-associated sites. Colors indicate fold change relative to 2xY->F controls.

E, F   Heatmap showing phosphosites from CD33, Siglec-5, and Siglec-8 overexpression that were (E) increased or (F) decreased with Siglec expression and shared their directional change with Siglec-F. Siglec-F column indicates average of all untreated Siglec-F replicates. Color scale is same as (D).

G   Western blot quantification of myc, SHP-2, and SHP-1 from myc co-IP eluates. Co-IP lysates were prepared from BV-2 cells stably expressing Siglec-F and optionally treated with 1 µM SHP099.

H   Heatmap showing phosphosites that were perturbed by SHP099 treatment. Color scale is same as (D).

role in inhibiting selected microglial signaling events while activating inflammatory response pathways.

Phosphorylation of Siglec ITIM sites is known to recruit SHP-2, a tyrosine phosphatase linked to Erk1/2 activation (Chemnitz *et al*, 2004; Sang *et al*, 2019). SHP099 is a recently developed allosteric inhibitor of SHP-2 (Chen *et al*, 2016). By pulling down against a C-terminal myc epitope tag on Siglec-F, we found that SHP099 disrupted the interaction between SHP-2 and Siglec-F at 1 μM without blocking SHP-1 (Fig 5G). To understand the role of SHP-2 in mediating signaling from the ITIM phosphorylation sites, we performed quantitative pTyr analysis of SHP099-treated cells. Out of 41 unique pTyr sites upregulated by Siglec-F expression, 7 were decreased with SHP-2 inhibition in this analysis (Fig 5H). These changes included SHP-2 (Ptpn11), as well as proteins involved in endocytosis (Pag1, Cbl) and immune processes (Lyn, Triobp, Fyb1, Dapp1). Thus, SHP-2 acts as one factor modulating Siglec-F-mediated pTyr signaling activation. Other ITIM-binding factors such as SHP-1 (Ptpn6), Inpp5d/Inppl1, or Pik3r1/2 may also mediate the effects of Siglec-F on pTyr signaling (Sweeney *et al*, 2005; Liu *et al*, 2015).

### Siglecs activate endocytic and phagocytic responses

We next assessed whether Siglec-F expression could alter microglial function, including endocytosis and phagocytosis. Previous literature has shown that CD22 and CD33 may alter microglial phagocytosis of protein aggregates and cellular debris (Griciuc *et al*, 2013, 2019; Pluvinage *et al*, 2019; Bhattacherjee *et al*, 2019). In contrast, Siglec-F and Siglec-8 have been shown to regulate endocytosis and vesicular trafficking in other cellular contexts (Walter *et al*, 2007; Tateno *et al*, 2007; Siddiqui *et al*, 2017; O'Sullivan *et al*, 2018). We tested this endocytic phenotype by co-culturing BV-2 cells that stably express Siglec-F together with empty vector control cells for 24 h. Cells were then incubated with Fluor 488-conjugated monomeric Aβ$_{1-42}$ and analyzed using flow cytometry (Fig 6A). We found that increased Siglec-F expression resulted in Aβ uptake compared to control cells (Fig 6B). In contrast, Siglec-F 2xY->F expression did not alter Aβ uptake (Fig 6C). We quantified relative fluorophore uptake between Siglec$^+$ and empty vector cells and found that overexpression of Siglec-F, CD33, Siglec-5, and Siglec-8 could all increase cellular uptake of monomeric Aβ$_{1-42}$ (Fig 6D). We saw a similar effect using 10 kDa dextran, suggesting this effect is not specific to Aβ (Fig 6E). Finally, we tested 1 μm fluorescent beads which can differentiate small scale endocytic events from phagocytosis. We found that overexpression of all four Siglecs could increase bead uptake, while 2xY->F showed the same level of uptake as empty vector (Fig 6F).

To show that these molecules were not binding to the surfaces of cells, we examined the uptake of pHrodo-tagged Dextran which fluoresces only at low pH within cellular lysosomes. BV-2 cells stably expressing Siglec-F showed increased pHrodo fluorescence intensity relative to 2xY->F cells (Fig 6G). Using this system, we found that SHP-2 inhibitor SHP099 was able to inhibit pHrodo Dextran uptake at concentrations > 500 nM (Fig 6H). Up to 2.5 μM SHP099 did not have an effect on the growth rate of control cells expressing 2xY->F receptors, however it appeared to further decrease the proliferation of Siglec-F-expressing cells (Fig 6I). Together, these data and previous literature support the case that Siglec-F drives an endocytic response

via SHP-2 signaling, downregulating its own surface expression and trafficking substrates to endolysosomal vesicles for degradation.

## Discussion

### Cross-model phosphoproteomics link signaling changes to AD pathologies

In this study, we sought to identify signaling pathways that are dysregulated in various mouse models of neurodegeneration using quantitative phosphoproteomics. Using mass spectrometry, we have quantified the phosphoproteomes of CK-p25 (Cruz *et al*, 2003, 2006; Fischer *et al*, 2005), 5XFAD (Oakley *et al*, 2006), and Tau P301S mice (Yoshiyama *et al*, 2007). We used a sensitive sample enrichment strategy that allowed for quantitative analysis of the low-abundance pTyr peptides from hippocampal and cortical tissues. Extensive signaling alterations were observed in all three models, linking AD pathologies to specific signaling pathway changes. Most importantly, we identified a common response involving Siglec-F and Inpp5d that we predicted to be linked to microglia using published cell-type-specific RNA-seq (Zhang *et al*, 2014, 2016) and single-cell RNA-seq datasets (Keren-Shaul *et al*, 2017; Mathys *et al*, 2017).

### Siglec-F's role in regulating microglial activation

Our work here indicates that human Siglec-8 and mouse Siglec-F receptors are upregulated on a subset of reactive microglia in AD patients and across various models of neurodegeneration. Using an integrative phosphoproteomics approach, we identified phosphorylation on Siglec-F in three mouse models of neurodegeneration. We found that Siglec-F was upregulated on a subset of microglia at an early stage of disease progression in CK-p25 and 5XFAD mice. The human paralog, Siglec-8, was also found to be upregulated in aged individuals and in patients with late-onset AD. Using *in vitro* models, we found that IFNγ stimulation upregulates Siglec-F and Siglec-8. These results support a model in which the expression of Siglec-F and Siglec-8 changes on microglia during neurodegeneration and aging, potentially altering microglial inflammatory activity.

One primary phenotype that we found to be associated with Siglec-F expression on BV-2 cells was the induction of cell death. We found that this effect was dependent on Siglec-F substrate-binding and ITIM phosphorylation. Our transcriptomic data suggest that Siglec-F triggers cell death via a pro-inflammatory pathway involving activation of the inflammasome and pyroptosis. Previous work on Siglec-F and Siglec-8 has shown that antibody engagement of these receptors causes a pro-apoptotic response in eosinophils, making them a potential drug target for asthma (Zimmermann *et al*, 2008; Kiwamoto *et al*, 2012). In contrast, genetic knock-out of Siglec-F leads to increased inflammation in asthma due to impaired apoptosis in eosinophils expressing Siglec-F (Kiwamoto *et al*, 2015; Engblom *et al*, 2017; Aran *et al*, 2019). One unanswered question in defining the role of Siglec-F and Siglec-8 in neurodegeneration is whether this pyroptotic response is acting as a "brake" on neuroinflammation by reducing the number of microglia or if it is adding fuel to the fire by increasing pro-inflammatory cytokine production. The development of brain-penetrant antibodies or bioconjugates

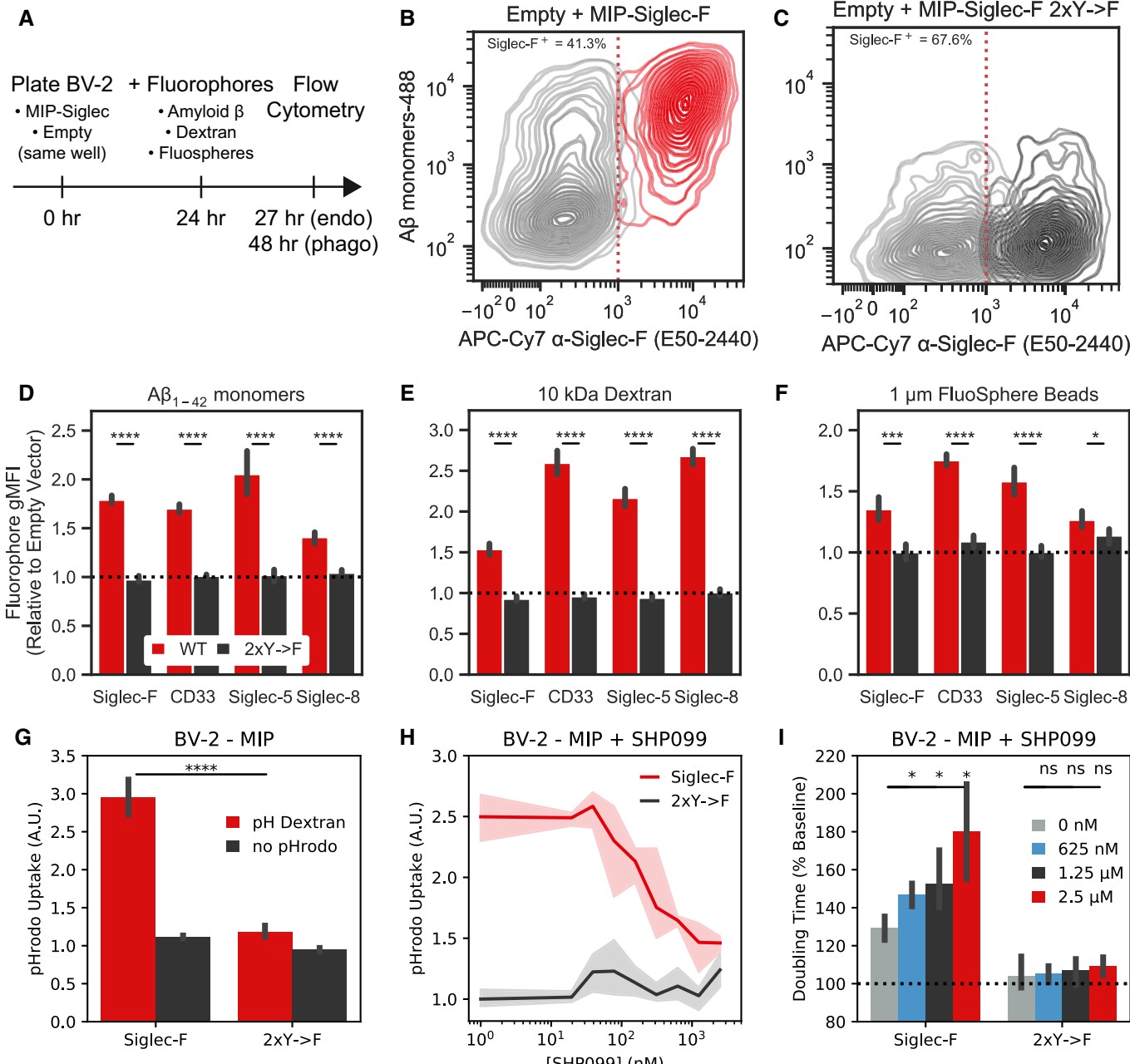

**Figure 6. Siglec-F expression drives an endocytic response in BV-2 cells.**

A Schematic showing experimental setup to measure rates endocytosis in BV-2 cells. Control (MIP empty vector) cells were plated together with cells stably expressing different Siglec constructs. Cells are allowed to adapt to the same media environment for 24 h, and then, fluorophores are added to media. Cells with endocytosis substrates (Aβ, Dextran) were assayed after 3 h, while phagocytosis substrates (FluoSpheres) were assayed after 24 h. Populations are separated, and relative uptake is measured using flow cytometry.

B, C Flow cytometry quantification of fluorescent monomeric Aβ uptake and Siglec-F expression for (B) Siglec-F and (C) Siglec-F 2xY->F. Siglec-expressing cells were plated with empty vector control cells in the same well alongside fluorescent substrates.

D–F Quantification of relative mean fluorescence intensities (MFI) between empty vector and Siglec-expressing populations. Relative values are shown for each mouse and human Siglec construct for uptake of (D) monomeric Aβ, (E) 10,000 MW Dextran, and (F) 1 μm FluoSpheres. Bars indicate mean ± 95% CI; n = 6 replicates; $^*P < 5e\text{-}2$, $^{**}P < 1e\text{-}2$, $^{***}P < 1e\text{-}3$, $^{****}P < 1e\text{-}4$; ns: not significant, using unpaired Student's *t*-test, two-sided. Dotted line indicates mean of 2xY->F group.

G pHrodo Dextran uptake in BV-2 cells with stable expression of Siglec-F estimated by Incucyte measurements. Bars indicate mean ± 95% CI; n = 6 replicates; $^*P < 5e\text{-}2$, $^{**}P < 1e\text{-}2$, $^{***}P < 1e\text{-}3$, $^{****}P < 1e\text{-}4$; ns: not significant, using unpaired Student's *t*-test, two-sided.

H pHrodo dextran uptake in BV-2 cells stably expressing Siglec-F with 0 - 2.5 μM SHP099 treatment. Data is from the same experiment as (G).

I Confluency doubling times for BV-2 cells with stable expression of Siglec-F with 0 - 2.5 μM SHP099 treatment. Data is from the same experiment as (G). Bars indicate mean ± 95% CI; n = 4 replicates; $^*P < 5e\text{-}2$, $^{**}P < 1e\text{-}2$, $^{***}P < 1e\text{-}3$, $^{****}P < 1e\text{-}4$; ns: not significant, using unpaired Student's *t*-test, two-sided. Dotted line indicates mean of untreated 2xY->F group.

targeting Siglec-F and Siglec-8 may allow for acute depletion of reactive microglia. Our work here suggests also that targeting signaling pathways downstream of Siglec-F may reduce neuroinflammation associated with chronic receptor activation.

Previous works characterizing the genetics of Siglecs suggest that an AD-protective allele in CD33 improves the microglial response during neurodegeneration (Naj *et al*, 2011; Griciuc *et al*, 2013; Bradshaw *et al*, 2013; Chan *et al*, 2015; Siddiqui *et al*, 2017; Griciuc *et al*, 2019). CD22 was also identified as an anti-phagocytic receptor that was upregulated on aging microglia (Pluvinage *et al*, 2019). Other than CD22 and CD33, there has not yet been a careful analysis of the effects of other Siglec receptors on microglia. Genetic studies of Siglec-8 have identified several variants associated with higher risk for asthma (rs36498, rs10409962, and rs11672925), but these mutations have not yet been linked to microglial function (Gao *et al*, 2010; Angata, 2014). Beyond these three genes, there remains many open questions about the Siglec family. Do these receptors play redundant roles? What ligands does each receptor respond to? And how do each of these responses fit into the larger picture of Alzheimer's disease progression? By answering these questions, we may be able to improve future therapeutic design strategies targeting Siglec function.

# Materials and Methods

## Reagents and Tools table

| Reagent/Resource | Reference or Source | Identifier or Catalog Number |
|---|---|---|
| **Experimental Models** | | |
| HEK 293T cells (H. sapiens) | Forest White's lab | ATCC CRL-3216 |
| BV-2 cells (M. musculus) | Li-Huei Tsai's lab | CVCL_0182 |
| AG09173 APOE3 induced-pluripotent stem cell line (H. sapiens) | Li-Huei Tsai's lab | N/A |
| CK-p25 (M. musculus) | Li-Huei Tsai's lab | Stock NO: 005706 |
| 5xFAD (M. musculus) | Jackson laboratories | Stock NO: 34840 |
| Δp35KI (M. musculus) | Li-Huei Tsai's lab | Stock NO: 022401 |
| Tau P301S (M. musculus) | Jackson laboratories | Stock NO: 008169 |
| C57BL/6 (M. musculus) | Jackson laboratories | Stock NO: 664 |
| Stbl3 | Thermo Fisher Scientific | C737303 |
| Human AD brain samples | Banner Sun Health Research Institute | N/A |
| **Recombinant DNA** | | |
| pCMV-VSV-G | Richard Hyne's lab | N/A |
| pUMVC | Richard Hyne's lab | N/A |
| psPAX2 | Addgene | 12260 |
| pMD2.G | Addgene | 12259 |
| MSCV-IRES-puro (MIP) | Richard Hynes's lab | N/A |
| pINDUCER20 | Addgene | 44012 |
| MIP-Siglec-F-myc | This paper | N/A |
| MIP-Siglec-F-Y538F-myc | This paper | N/A |
| MIP-Siglec-F-Y561F-myc | This paper | N/A |
| MIP-Siglec-F-2xY->F-myc | This paper | N/A |
| MIP-hCD33-myc | This paper | N/A |
| MIP-hCD33-myc | This paper | N/A |
| MIP-hSiglec-5-myc | This paper | N/A |
| MIP-hSiglec-5-2xY->F-myc | This paper | N/A |
| MIP-hSiglec-8-myc | This paper | N/A |
| MIP-hSiglec-8-2xY->F-myc | This paper | N/A |
| pINDUCER20-Siglec-F-myc | This paper | N/A |
| pINDUCER20-Siglec-F-dVset-myc | This paper | N/A |

**Reagents and Tools table**  (continued)

| Reagent/Resource | Reference or Source | Identifier or Catalog Number |
|---|---|---|
| pINDUCER20-Siglec-F-2xY->F-myc | This paper | N/A |
| **Antibodies** | | |
| Anti-phospho-Tyr (4G10) | Millipore | 05-321 |
| Anti-phospho-Tyr (PT66) | Sigma | P3300 |
| Anti-phospho-Ser/Thr-Pro (MPM-2) | Millipore | 05-368 |
| Anti-MAPK/CDK Substrate (34B2) | Cell Signaling Technology | 2325 |
| Anti-Iba1 (Polyclonal) | Synaptic Systems | 234 004 |
| Anti-Siglec-F (E50-2440) | BD Biosciences | 552125 |
| Anti-myc (9B11; Sepharose Bead Conjugate) | Cell Signaling Technology | 3400 |
| Anti-myc (9B11) | Cell Signaling Technology | 9B11 |
| Anti-myc (71D10) | Cell Signaling Technology | 2278 |
| Anti-Siglec-8 (ab38578) | Abcam | ab38578 |
| Anti-Amyloid Beta (D54D2) | Cell Signaling Technology | 8243 |
| Anti-MHC-II (HLA-DP, HLA-DQ, HLA-DR; CR3/43) | Agilent | M077501-2 |
| Anti-SHP-1 (C14H6) | Cell Signaling Technology | C14H6 |
| Anti-SHP-2 (D50F2) | Cell Signaling Technology | D50F2 |
| APC-Cy7 Anti-Siglec-F (E50-2440) | BD Biosciences | 565527 |
| APC Anti-CD33 (WM53) | BD Biosciences | 561817 |
| Alexa-647 Anti-Siglec-5/14 (194128) | BD Biosciences | 564371 |
| APC Anti-Siglec-8 (7C9) | Biolegend | 347105 |
| Anti-IL-1β (Polyclonal) | R&D Systems | AF-401-SP |
| Anti-β-tubulin (Polyclonal) | Cell Signaling Technology | 2146 |
| **Oligonucleotides and sequence-based reagents** | | |
| qPCR Primers | This Study | Dataset EV6 |
| **Chemicals, enzymes and other reagents** | | |
| PBS | Thermo Fisher Scientific | 10010023 |
| Formaldehyde | Sigma | 252549 |
| Urea | Sigma | U5128 |
| Pierce BCA Protein Assay | Thermo Fisher Scientific | 23225 |
| Dithiothreitol (DTT) | Sigma | D0632 |
| Iodoacetamide (IAA) | Sigma | I1144 |
| Sequencing Grade Modified Trypsin | Promega | V5111 |
| 99.99% Acetic Acid | Sigma | 338826 |
| Sep-Pak Lite C18 Cartridge | Waters | WAT023501 |
| Sep-Pak Plus C18 Cartridge | Waters | WAT020515 |
| Pierce Quantitative Colorimetric Peptide Assay | Thermo Fisher Scientific | 23275 |
| 6-plex Tandem Mass Tag | Thermo Fisher Scientific | 90061 |
| 10-plex Tandem Mass Tag | Thermo Fisher Scientific | 90110 |
| ZORBAX 300Extend-C18 | Agilent | 770995-902 |
| High-Select™ Fe-NTA Phosphopeptide Enrichment Kit | Thermo Fisher Scientific | A32992 |
| 10 μm C18 beads | YMC | ODS-A AA12S11 |
| 5 μm C18 beads | YMC | ODS-AQ AQ12S05 |
| Poros 20 MC Metal Chelate Affinity Packing | Poros | 1-5429-06 |

**Reagents and Tools table** (continued)

| Reagent/Resource | Reference or Source | Identifier or Catalog Number |
|---|---|---|
| Fused Silica Capillary Tubing 50 μm ID | Polymicro Technologies | 1068150017 |
| Fused Silica Capillary Tubing 100 μm ID | Polymicro Technologies | 1068150023 |
| Fused Silica Capillary Tubing 200 μm ID | Polymicro Technologies | 1068150204 |
| Neuraminidase (Sialidase) from Arthrobacter ureafaciens | Sigma | 10269611001 |
| FuGene | Promega | E2311 |
| PEI MAX | Polysciences Inc | 24765-1 |
| .45 μm syringe filters | Sigma | 431220 |
| BD syringe | VWR | 302995 |
| RPMI 1640 Medium, GlutaMAX Supplement | Thermo Fisher Scientific | 61870036 |
| DMEM media | VWR | 10013CV |
| Fetal Bovine Serum (FBS), Certified | Thermo Fisher Scientific | 16000-044 |
| Pen Strep | Thermo Fisher Scientific | 15140-122 |
| 0.05% Trypsin | Thermo Fisher Scientific | 25300-120 |
| Annexin V binding/washing buffer | Thermo Fisher Scientific | V13246 |
| Annexin V-488 | Thermo Fisher Scientific | A13201 |
| Propidium Iodide | Thermo Fisher Scientific | P1304MP |
| PNGase F | New England BioLabs | P0704S |
| pHrodo Green Dextran | Thermo Fisher Scientific | P35368 |
| Superscript IV VILO Master Mix | Thermo Fisher Scientific | 11756050 |
| Ultrapure Water for HPLC | Sigma | 270733-4L |
| iQ SYBR Green Supermix | Biorad | 1708882 |
| Microseal 384-well PCR Plate | Biorad | MSP3842 |
| SHP099 | SelleckChem | S8278 |
| Bestatin | Sigma | B8385 |
| Tofacitinib citrate | Sigma | PZ0017 |
| Sodium Citrate | Sigma | S8750 |
| Hoechst 33342 | Thermo Fisher Scientific | H3570 |
| Matrigel | Corning | BD354277 |
| ReLeSR | StemCell Technologies | 5872 |
| mTESR | StemCell Technologies | 85850 |
| STEMdiff Hematopoietic Kit | StemCell Technologies | 5310 |
| ROCK inhibitor (Y-27632 dihydrochloride) | Tocris | 1254 |
| DMEM/F12 | Thermo Fisher Scientific | 11330-057 |
| ITS-G | Thermo Fisher Scientific | 41400045 |
| B27 | Thermo Fisher Scientific | 17-504-044 |
| N2 | Thermo Fisher Scientific | 17502048 |
| Glutamax | Thermo Fisher Scientific | 35050061 |
| MEM Non-essential Amino Acid (NEAA) Solution | Sigma | M7145-100ML |
| Penicillin:Streptomycin Solution | Gemini Bio-Products | 400-109 |
| Insulin | Sigma | 91077C-250MG |
| Monothioglycerol | Sigma | M6145-100ML |
| M-CSF | PeproTech | 100-21-1MG |
| IL-34 | PeproTech | 300-25-1MG |
| TGFβ-1 | PeproTech | 200-34-500UG |

 

**Reagents and Tools table** (continued)

| Reagent/Resource | Reference or Source | Identifier or Catalog Number |
|---|---|---|
| CD200 | Novoprotein | C311 |
| CX3CL1 | PeproTech | 300-31-1MG |
| Human IFN-α2 | StemCell Technologies | 78076 |
| Human IFN-β | R&D Systems | 8499-IF |
| Human IFN-γ | R&D Systems | 285-IF |
| Mouse IFN-γ | VWR | 575302-BL |
| TRIzol | Thermo Fisher Scientific | 15596018 |
| Chloroform | Sigma | C2432 |
| Ethanol, 200 Proof | VWR | V1001 |
| RIPA Buffer (2X) | Boston BioProducts | BP-115X |
| Halt Protease and Phosphatase Inhibitor Cocktail | Thermo Fisher Scientific | 1861281 |
| LDS Sample Buffer | Invitrogen | NP0007 |
| SDS-PAGE gel | Invitrogen | NP0335BOX |
| Nitrocellulose Membranes, 0.2 μm | Bio-Rad | 1620112 |
| Immun-Blot PVDF Membrane | Bio-Rad | 162-0177 |
| Novex Tris-Glycine Transfer Buffer | Thermo Fisher Scientific | LC3675 |
| Tris-Buffered Saline | Corning | 46-012-CM |
| Tween 20 | Fisher BioReagents | BP337 |
| Intercept Blocking Buffer | Li-Cor | 927-70001 |
| TrueBlack Lipofuscin Autofluorescence Quencher | Biotium | 23007 |
| Direct-zol RNA MicroPrep Kit | Zymo Research | R2061 |
| Dynabeads mRNA Direct Kit | Thermo Fisher Scientific | 61012 |
| Kapa mRNA Hyperprep Kit | Roche | 8098115702 |
| pENTR/D-TOPO Cloning Kit | Thermo Fisher Scientific | K240020 |
| Gateway LR Clonase | Thermo Fisher Scientific | 11791020 |
| Software | | |
| Proteome Discoverer (v2.2.1) | Thermo Fisher Scientific | https://www.thermofisher.com/us/en/home/industrial/mass-spectrometry/liquid-chromatography-mass-spectrometry-lc-ms/lc-ms-software/multi-omics-data-analysis/proteome-discoverer-software.html |
| MASCOT (v2.4.1) | Matrix Science | http://www.matrixscience.com/ |
| XCalibur (v2.2) | Thermo Fisher Scientific | https://www.thermofisher.com/order/catalog/product/OPTON-30965 |
| IncuCyte ZOOM (v2016A) | Sartorius | https://www.essenbioscience.com/en/products/incucyte-zoom-resources-support/software-modules-incucyte-zoom/ |
| STAR (v2.5.3a) | (Dobin et al, 2013) | https://github.com/alexdobin/STAR/releases |
| RSEM (v1.3.0) | (Li & Dewey 2011) | http://deweylab.github.io/RSEM/ |
| DESeq2 (v1.18.1) | Bioconductor | https://www.bioconductor.org/packages/devel/bioc/html/DESeq2.html |
| GSEA (v3.0 beta-2) | Mootha et al. (2003), Tamayo et al (2005) | https://www.gsea-msigdb.org/gsea/ |
| R (v3.4.4) | N/A | https://www.r-project.org/ |
| Salmon (v0.9.1) | Patro, Duggal, Love, Irizarry, and Kingsford (2017) | https://combine-lab.github.io/salmon/ |
| Python (v3.7.1) | N/A | https://www.python.org |

**Reagents and Tools table**  (continued)

| Reagent/Resource | Reference or Source | Identifier or Catalog Number |
| --- | --- | --- |
| MAGIC (v1.5.5) | van Dijk *et al* (2018) | https://www.krishnaswamylab.org/projects/magic |
| pyproteome (v0.11.0) | This publication | https://github.com/white-lab/pyproteome |
| scikit-learn (v0.21.1) | Pedregosa *et al* (2012) | https://scikit-learn.org/stable/ |
| scipy (v1.3.1) | Virtanen *et al* (2020) | https://www.scipy.org/ |
| numpy (v1.17.0) | Van Der Walt, Colbert, and Varoquaux (2011) | https://numpy.org/ |
| pandas (v0.25.0) | McKinney (2010) | https://pandas.pydata.org/ |
| seaborn (v0.10.0) | Waskom (2018) | http://seaborn.pydata.org/ |
| matplotlib (v3.1.1) | Hunter (2007) | http://matplotlib.org/ |
| goenrich (v1.12) | Rudolph *et al* (2016) | https://github.com/jdrudolph/goenrich |
| goatools (v1.0.3) | Klopfenstein *et al* (2018) | https://github.com/tanghaibao/goatools |
| Cytoflow (v1.0) | N/A | https://bpteague.github.io/cytoflow/ |
| MSigDB (v6.1) | Liberzon *et al* (2015), Subramanian *et al* (2005) | http://software.broadinstitute.org/gsea/msigdb/ |
| Fiji-ImageJ (v1.52u) | National Inst. Of Health | https://imagej.net/Fiji |
| ZEN (v2.1 SP3) | Zeiss | https://www.zeiss.com/microscopy/us/products/microscope-software/zen.html |
| **Other** | | |
| Q Exactive Plus | Thermo | IQLAAEGAAPFALGMBDK |
| Q Exactive HF-X | Thermo | IQLAAEGAAPFALGMBFZ |
| LTQ Orbitrap | Thermo | N/A |
| VT100S vibratome | Leica | N/A |
| LSM 710 | Zeiss | N/A |
| LSM 880 | Zeiss | N/A |
| BD FACS Canto Cell Sorter | BD Biosciences | N/A |
| Li-Cor Odessey CLx | LI-COR Biosciences | N/A |
| Incucyte Plate Imager | Essen Bioscience | N/A |
| Illumina HiSeq 2000 | Illumina | N/A |
| CFX384 Touch Real-Time PCR | Biorad | N/A |

## Methods and Protocols

### Mouse models

All animal work was approved by the Committee for Animal Care of the Division of Comparative Medicine at the Massachusetts Institute of Technology. The mouse models used in this study were: (i) double transgenic CK-p25, CK control (aged 3–4 months, induced for 2–7 weeks by removal of doxycycline from diet); (ii) 5XFAD; 5XFAD Δp35KI$^{+/+}$, WT, WT Δp35KI$^{+/+}$ (9–11 months); and (iii) Tau P301S Tg and WT (4–9 months). Mice were housed in groups of three to five on a standard 12 h light/12 h dark cycle, and all experiments were performed during the light cycle. Food and water were provided ad libitum. Only male mice were used for the phosphoproteomics analyses with the exception of CK/CK-p25 mice that were induced for 7wk. Both male and female mice were used for IF experiments. The full list of each mouse ID, sex, genotype, and age used for proteomics is listed in Dataset EV1.

### Proteomics sample processing

Mice were deeply anesthetized in 5% isoflurane after which their cerebral cortex, cerebellum, and hippocampus were dissected while immersed in ice-cold PBS. Tissues were snap-frozen in liquid nitrogen and stored at −80°C. Tissues for proteomics analysis were homogenized in 3 ml ice-cold 8 M urea. Lysates were then centrifuged at 4000 g for 30 min to clear lipids and DNA. Aliquots were taken for a BCA assay of protein concentration. Lysates were then treated with 10 mM DTT for 1 h at 56°C, followed by 55 mM iodoacetamide for 1 h at room temperature, rotating in the dark. Samples were diluted by adding 8 ml Ammonium Acetate pH 8.9 and then digested at room temperature overnight on a rotator with trypsin at a 1:50 ratio of trypsin to total protein mass (1–2 mg total protein expected for hippocampus, 3–4 mg for cerebellum, and 8–10 mg for cortex). This reaction was quenched with 1 ml 99.99% Acetic Acid and samples were desalted using Sep-Pak Lite C18 cartridges for hippocampus and Sep-Pak Plus for cortex and cerebellum samples. Peptides were then dried to half volume in a speed-vac and post-cleanup concentrations

were assayed using a Pierce Quantitative Colorimetric Peptide Assay. Peptides were then divided into 130 μg aliquots that were snap-frozen and lyophilized. Peptides were then labeled with 6-plex and 10-plex isobaric tandem mass tags (TMT) according to manufacturer's protocol. Labeled samples were then dried down in a speed-vac overnight. Two or more technical replicates were run for each collection of tissues to improve pTyr coverage.

Phosphotyrosine analysis of BV-2 cells was performed by first transducing cells with MSCV-IRES-Puro retrovirus to generate stable Siglec expression lines. 4 days after transduction, cells were split into 10-cm dishes with 1 million cells/dish in 10 ml of fresh RPMI media. 1 μM SHP099 was added to cells 24 h before lysis. Cells were lysed after 48 h in ice-cold 8 M urea. Peptides with TMT 10-plex labels were prepared following the same protocol as mouse brain tissue above.

### Phosphopeptide enrichment and LC-MS/MS analysis

Tyrosine phosphorylated peptides were enriched by a two-step process consisting of an immunoprecipitation (IP) with two pan-specific anti-phosphotyrosine antibodies (4G10, PT66) followed by immobilized metal affinity chromatography (IMAC) as previously described (Wolf-Yadlin *et al*, 2006; Johnson & White, 2012; Johnson *et al*, 2012; Gajadhar *et al*, 2015; Reddy *et al*, 2016). IP supernatants were subjected to a second round of IP with anti-pSer/pThr motif antibodies (MPM-2; 34B2) followed by IMAC cleanup step. IP supernatants were then divided into 80 fractions using high pH reverse-phase chromatography on a ZORBAX C18 column. Fractions were concatenated into 20 tubes and dried down. Each fraction was then enriched using commercial Fe-NTA columns. Small amounts of each IP supernatant were saved and diluted 1:1,000 in 0.1% acetic acid for global protein expression profiling.

Phosphopeptide-enriched samples were loaded onto a BSA-conditioned pre-column with 10 μm C18 beads. Columns were rinsed with 0.1% acetic acid to remove excess salts and attached to an analytical column with 10 cm of 5 μm C18 beads. Agilent 1100 Series HPLCs were operated at 0.2 ml/min flow rates with a pressure-restricted T-junction to attain nanoliter flow rates. Peptides were eluted with increasing concentrations of buffer B (70% acetonitrile, 0.1% acetic acid) using the gradient settings: 0–13% 0% 8 (0 min), 13% 8 (10 min), 42% 8 (105 min), 60% 8 (115 min), 100% 8 (122 min), 100% 8 (128 min), 0% 8 (130 min). Global phosphoproteome and proteome fractions were analyzed using an EASY-nLC nano-flow UHPLC. Fractions were eluted using the gradient settings: 0–10% 0% 8 (0 min), 10% 8 (10 min), 30% 8 (110 min), 40% 8 (124 min), 60% 8 (129 min), 100% 8 (131 min), 100% 8 (141 min), 0% 8 (143 min). Peptides were ionized by electrospray ionization (ESI) at 1–3 kV.

Peptides were then analyzed by LC-MS/MS on a QExactive Plus and QExactive HF-X Orbitrap mass spectrometer operating in data-dependent mode acquiring MS scans and HCD MS/MS fragmentation spectra. Isolation width was set to 0.4 Da (full width) to decrease isolation interference and TMT ratio compression. Ions with charge> 1 were dynamically selected for fragmentation using a top-20 untargeted method with an exclusion time of 30s. The maximum injection time and ACG targets were set to 50 ms and 3e6 respectively. MS scans were captured with resolution = 60,000 and MS2 scans with resolution = 45,000. Peptides were fragmented with

the HCD normalized collision energy set to 33%. Protein expression profiling was performed on LTQ Orbitrap or QExactive Plus instruments.

### Proteomics data processing and integration

Peptide-spectrum matching and quantification was run using Proteome Discoverer and MASCOT. Peptides were searched against the UniProt mouse proteome database (UP000000589, SwissProt Version 2020_01) with a max missed tryptic cleavage count of 2. Tau P301S mice and 5XFAD were also searched against a secondary database containing transgenic human MAPT (isoform E, P301S), APP (isoform C K670N/M671L; I716V; V717I), and PSEN1 (isoform I-463 M146L; L286V) sequences. Precursor and fragment ions were matched with 10 ppm and 20 mmu mass tolerances, respectively. 0.8 Da mass tolerances were used for samples run on the LTQ Orbitrap. Variable modifications were included for phosphorylation (STY), oxidation (M), and TMT (N-term, K). Fixed modifications were set for carbamidomethyl (C). Phosphorylation sites were localized using the ptmRS module with a confidence threshold of 0.75. PSMs for Siglec-F pY561 (SVyTEIK) were manually added to data frames after manually validating spectrum with the correct mass and retention time (m/z = 689.37, z = 2, RT = 55–85 min). TMT peaks for these spectra were manually quantified using Xcalibur Qual Browser to extract peak heights. We compared TMT quantification values for scans that were matched by Proteome Discoverer and observed < 5% difference between the two methods, indicating that manual TMT quantification is generally accurate. Phosphosites for Ptpn11 (Y62), Lyn (Y508), Fgr (Y511), Fyn (Y531), and Siglec-F (Y561) were reassigned after manual validation of spectra with an updated version of CAMV (Curran *et al*, 2013). Schematic images in Figs 1A and 5A, and the synopsis image were created were created with BioRender (https://biorender.com).

Searched '.msf' files were imported and processed using pyproteome, an in-house tool for phosphoproteomics data integration and analysis. Peptide-spectrum matches (PSMs) were read directly into pandas data frames from Proteome Discoverer '.msf' files using programmed SQLite queries. PSMs were filtered using an ion score cutoff> 15, isolation interference < 30, and median TMT signal> 1500, and Percolator FDR < 1e-2. Transgenic peptides were excluded from filtering on percolator values. TMT quantification data were normalized using a iterative fitting procedure based on the CONSTrained STANdardization algorithm (Maes *et al*, 2016). This procedure iteratively adjusts the matrix of filtered TMT quantification values such that rows were mean-centered around 1 and columns were mode-centered around 1. Mode centers were calculated by fitting a gaussian KDE distribution (scipy.stats.kde.gaussian_kde) to column intensities and finding the max value of its probability density function. Duplicate PSMs were then grouped for final quantification using a weighted mean function: $TMT * (1 - \frac{\text{isolation inference}}{100})$ for the TMT intensities and isolation interference value quantified by Proteome Discoverer for each PSM. Heatmaps were generated using seaborn and matplotlib.

### Motif enrichment analysis

Motif logo enrichment values were calculated according to the method described for pLogo (O'Shea *et al*, 2013). Only non-ambiguous phosphopeptides (ptmRS probability > 0.75) were analyzed,

each phosphorylated site on a peptide was considered as its own N-mer. N-mers were set to 15 amino acids. Foreground and background N-mers were extracted from protein-aligned sequences, using "A" to pad N-and C-terminal peptides. Quantification values from identical N-mers (mis-cleaved, oxidized, or multiply phosphorylated peptides) were combined by taking the median value. Log odds enrichment values are equal to $-\log_{10} \frac{sf(k-1)}{cdf(k)}$, with k = the count of amino acids at that relative position for phosphosites in a foreground collection ($P < 0.05$; FC > 1.25 or FC < 0.8). Survival function (sf) and cumulative distribution functions (cdf) were calculated using scipy's hypergeometric distribution model (scipy.stats.hypergeom), with: $n$ = the total number of foreground N-mers; K = the count of amino acids at that position in the background collection; and $N$ = the total number of background N-mers.

The number of residues matching a phosphorylation motif was calculated for each phosphorylated site on a peptide and reported as the maximum number for multiply phosphorylated peptides. The patterns used were: (i) CaMKII Motif: - [FLMVI] - [RK] - Q -. - [st] - [FLMVI]-; (ii) CDK Motif: - [st] - P - K - K -. For this pattern, "." matches any residue and [st] matches pSer/pThr. Wildcards and phosphosites were not included in the total residue count for motifs.

### Cell-type enrichment analysis

Gene cell-type predictions were estimated using a cell-type-specific sequencing atlas (Zhang et al, 2014, 2016). Cell expression values were extracted using the following columns: "Astrocyte": ["1 month", "4 months", "7 months", "9 months"], "Neuron": ["Neuron 3", "Neuron 4"], "OPC": ["Oligodendrocyte precursor cell 3", "Oligodendrocyte precursor cell 4"], "New Oligodendrocytes": ["Newly formed oligodendrocyte 3", "Newly formed oligodendrocyte 4"], "Myelinating Oligodendrocytes": "Myelinating oligodendrocyte 4", "Myelinating oligodendrocyte 5"], "Microglia": ["Microglia 1", "Microglia 2"], "Endothelia": ["Endo 1", "Endo 2"]. Genes were calculated as being enriched in a given cell type if they met the criteria $\frac{\mu_{celltype}}{\sum_{i \neq celltype} \mu_i} > 2.5$, where μ is the mean FPKM value from collections of columns for each cell type. OPC and New Oligodendrocytes were excluded from the enrichment calculation to avoid exclusion of pan-oligodendrocyte genes, only "Myelinating Oligodendrocytes" was used for the displayed "Oligodendrocyte" category.

Cell-type enrichments were calculated on peptide clusters defined by upregulation ($P < 1e-2$, FC > 1.25) or downregulation ($P < 1e-2$, FC < 0.8). Peptides were compared using pooled data from the cortex and hippocampus from all mice with each diseased genotype compared to its WT controls. Peptides in clusters mapping to only one protein were processed into a list of unique genes with at least one PSM (modified or unmodified) that were enriched in any cell type. Log odds enrichment (LOE) values were calculated from hypergeometric distribution functions calculated from foreground and background sets in a manner similar to motif logos above.

### Phosphosite enrichment and pathway analysis

Phosphosite enrichment analysis (PSEA) were performed using the procedure outlined in a previous publications (Krug et al, 2018). We used a custom phosphosite database derived from Phosphosite Plus (PSP; Hornbeck et al, 2015). Kinase-substrate mappings were downloaded from PSP using information for all species (Kinase_Substrate_Dataset.gz), and then re-mapped to mouse

phosphosites using homology information from PSP (Phosphorylation_site_dataset.gz). Phosphopeptide fold changes were estimated from the median value of redundant peptides. Fold changes were z-scored and phosphosites were rank ordered. Enrichment scores were calculated using the integral of a running-sum statistic (exponential weight = 0.75) that was increased for each gene or site contained within a given set and decreased for each gene not contained in that set. Enrichment scores for 1,000 matrices with scrambled rows were generated and used to calculate an empirical $P$-value, $q$-value, and normalized enrichment score (NES). Only phosphosite sets with a minimum overlapping set size of 20 were scored. Phosphosite sets that were enriched in at least two models were selected for display.

Gene ontology pathway analysis was performed using goenrich (Rudolph et al, 2016). Foreground and background protein IDs were generated from all phosphopeptides that were upregulated (FC > 1.25, $P < 1e-2$) or downregulated (FC < 0.8, $P < 1e-2$) in each mouse model comparison. Protein IDs were mapped to Entrez IDs and enrichment scores were calculated using the NCBI go-basic and gene2go databases. Gene ontologies were generated using goatools (Klopfenstein et al, 2018). The following GO terms were used for each category: Signal Transduction: GO:0016791 (phosphatase activity), GO:0007165 (signal transduction); Endosome: GO:0005768 (endosome); Adhesion: GO:0007155 (cell adhesion); Cytoskeleton: GO:0005856 (cytoskeleton); Metabolism: GO:0008152 (metabolic process).

### Human samples

Samples of fixed free-floating brain slice tissue were secured from AD and neurologically normal, non-demented elderly control (ND) brains obtained at autopsy at the Banner Sun Health Research Institute Tissue Bank (BSHRI). The brain bank at BSHRI is one of the world's best, dedicated to the highest standards (average RNA integrity; RIN 8.5) and postmortem interval (2.8 h) (Birdsill et al, 2011; Walker et al, 2016). Cognitive status of all cases was evaluated antemortem by board-certified neurologists, and postmortem examination by a board-certified neuropathologist resulting in a consensus diagnosis using standard NIH AD Center criteria for AD or ND. Samples analyzed in the current study were collected from 3 male and 5 female human subjects with ages ranging from 60–96 and 60–82 years, respectively. The full list of each ID, sex, age, and diagnostic histopathology measurements for patients is listed in Dataset EV3.

### Immunofluorescence and microglia imaging analysis

For hippocampus slice analysis, mice were perfused transcardially with PBS followed by 4% paraformaldehyde (PFA) to fix tissue. Brains were then dissected and post-fixed with 4% PFA overnight. The tissue was then sliced into 40 μm sections using a Leica VT100S vibratome and stored in PBS at 4°C. Antigen retrieval was performed on human fixed free-floating tissue by washing glycerol-preserved slices three times with TBST, boiling samples for 10 min at 95°C in 10 mM citric acid, 0.05% Tween 20, pH 6, and then washing twice with TBST. Brain slices were blocked and permeabilized in 10% Normal Donkey Serum (NDS), 0.2% Triton-X100 for 1 h. Blocked slices were then incubated with primary antibody for 1–2 days at 4°C. The following antibodies were used for primary staining: α-MHC-II (CR3/43; 1:100); α-Iba1 (polyclonal; 1:500); α-

Siglec-F (E50-2440; 1:100); α-Siglec-8 (ab198690; 1:50). Slices were washed three times with PBS with shaking for 5 min between washes. Slices were then stained with secondary antibody (donkey host, 1:500), and Methoxy-X04 (100 μM) to stain plaques or Hoechst 33342 (1:10,000) to stain nuclei. Slices were treated with 1X True-Black for 30 s after immunostaining and then washed 3 times in PBS to reduce autofluorescence signal. All tissue slices shown and analyzed were treated with TrueBlack with the exception of the tile scan shown in Fig EV4B.

Slices were imaged on a Zeiss 880 confocal microscope, using a 20× and 63× objective to generate z-stack images from regions of interest. Images were taken with excitation/ emission window filters set up to avoid signal bleed between secondaries (Blue: 405 nm, 410–515 nm; Green: 488 nm, 493–577 nm; Red: 561nm, 585–634 nm; Far-Red: 633 nm, 638–755 nm). Super-resolution images were taken using the AiryScan module through a 20× objective with 3–5× digital zoom.

For antibody validation, BV-2 cells transduced with pINDUCER20-Siglec constructs were plated in a 96-well with 500 ng/ml Doxycycline for 48 h and then fixed with 4% PFA for 15 min and washed with PBS. Cells were stained with α-Siglec-F (E50-2440; 1:100) or α-Siglec-8 (ab198690; 1:100) as well as Hoechst 33342 (1:10,000) to stain nuclei. Cells were imaged on a Zeiss 710 confocal microscope.

Z-stack projections, channel masks, area coverage, percentage overlap, and mask-proximity analyses were calculated using ImageJ. Z-stack slices were first normalized using the attenuation correction plugin (Biot et al, 2008). To remove background, mean slice signal intensities were multiplied by a constant factor (Siglec-8 = 3.5, Iba1 = 3, Siglec-F = 3.5, MHC-II = 3, Aβ = 8) and then subtracted from raw pixel intensities. Masks were generated from the Iba1, MHC, and Siglec channels using ImageJ's Huang method to automatically set thresholds, and then slices were filtered for 2d particles with size> 0.5 μm$^2$.

For overlap analysis, Siglec channel intensities were multiplied by copies of the Iba1 and/or MHC-II masks that were dilated in 3D by 2 voxels to cover neighboring antibody signals. Voxel mask binary operations were performed using the And/Or/Not operations provided by ImageJ's CLIJ plugin (Haase et al, 2020). Mask overlap was calculated as $\frac{Siglec \cap Iba1}{Siglec}$ for mask intensity and area values. Mask-proximity analyses were performed by calculating the distance between each voxel in an image mask and the nearest Aβ plaque (> 10 μm$^3$). Aβ plaques were located using ImageJ's 3D Objects Counter plugin. Proximity and mask overlap values were calculated for individual 20× wide-field images. Cumulative distribution functions, Kolmogorov–Smirnov (K-S), and 2-sample $t$-test statistics were generated using numpy and scipy (Van Der Walt et al, 2011; Virtanen et al, 2020).

## scRNA-Seq data analysis

Single-cell RNA-seq (scRNA-seq) data were downloaded for: (i) CK-p25 mice (Mathys et al, 2017) using NCBI:GEO accession GSE103334; (ii) 5XFAD mice (Keren-Shaul et al, 2017) using NCBI: GEO accession GSE98971. scRNA-seq datasets were smoothed for missing values using MAGIC (van Dijk et al, 2018). CK-p25 data were displayed using tSNE coordinates from the original publication. Cells clustered as "microglia" from 5XFAD and C57BL/6 mice in Keren-Shaul et al were re-projected using UMAP (McInnes et al, 2018) as original tSNE coordinates were not available.

## Cell culture models

Human embryonic kidney (HEK) 293T cells were from ATCC and cultured in DMEM containing 10% FBS and 1% Pen/Strep (Thermo). BV-2 cells were immortalized from a female C57BL/6 mouse background (Blasi et al, 1990) and cultured in RPMI 1640 medium with GlutaMAX supplemented with 10% FBS, and 1% Pen/Strep. BV-2 were maintained at ∼1–50% confluency and split every 3–4 days to avoid activation from overcrowding. All cells were incubated at 37°C with 5% CO$_2$. No mycoplasma contamination was detected in cell lines used in this study.

BV-2 cells stably expressing Siglec constructs were generated using retroviral vectors comprising a MSCV promoter and IRES-puro module (Hynes lab). Retroviral particles were packaged by transfecting 293T cells with MIP constructs, pCMV-VSV-G, and pUMVC packaging vectors. BV-2 cells were transduced for 6–8 h by co-incubation with 1 ml virus aliquots and 4 μg/ml polybrene. Transduced cells were selected after 24 h using 2 μg/ml puromycin for 2–3 days.

BV-2 cells with inducible Siglec expression were generated using lentiviral vectors comprising a TRE promoter alongside a UbC promoter expressing TetR (rtTA-Advanced) and IRES-neomycin resistance module (pINDUCER20) (Meerbrey et al, 2011). Lentiviral particles were packaged by transfecting 293T cells with pINDUCER constructs, psPAX2, and pMD2.G 2$^{nd}$ generation packaging vectors. BV-2 cells were transduced for 6–8 h by co-incubation with 1 ml virus aliquots. Transduced cells were selected after 24 h using 1 mg/ml G418 for 2–3 days.

## Induced pluripotent stem cell-derived Microglia (iMGLs)

iMGLs were generated from previously characterized APOE e3/e3 patient line (Lin et al, 2018) using a published protocol (Abud et al, 2017; McQuade et al, 2018). Induced pluripotent stem cells were grown in 6-wells on matrigel to 90% confluency and split with ReLeSR. Colonies of 50–200 cells (∼200 μm diameter) were counted and plated onto a matrigel-coated 6-well plate at a density of 100 clumps/well. The next day (day 0), media was changed to 2 ml/well of STEMdiff™ Hematopoietic Kit with Supplement A at 1:200. On day 2, 1 ml of STEMdiff™ with Supplement A was added to each well. On day 3, media was changed to 2 ml/well of STEMdiff™ with Supplement B. On days 5, 7, and 9, 1 ml/well of STEMdiff™ with Supplement B was added to each well. On day 11, non-adherent induced hematopoietic stem cells (iHPSCs) were collected from the media supernatant.

iMGLs were derived from iHPSCs by maturing cells in iMGL Differentiation Base Media (iMBM). iMBM was created by mixing 2x 500 ml DMEM/F12, HEPES (1:1), 20 ml ITS-G, 20 ml B27, 5 ml N2, 10 ml Glutamax, 10 ml NEAA, 10 ml Pen/Strep (Gemini), 500 μl insulin, and 35 μl monothioglycerol and straining through a .22 μm filter. iHPCs were spun at 300 $g$ for 5 min and re-plated in a 6-well on matrigel at a density of 200,000 cells/well in 2 ml iMBM with freshly added 25 ng/ml M-CSF, 100 ng/ml IL-34, and 50 ng/ml TGFβ-1. Freshly prepared iMBM was added to cells every other day from days 13–35. On days 23 and 36, media supernatant was collected, centrifuged at 300xg for 5 min, and resuspended in 1 ml fresh iMBM to reduce media volume without cell loss. On day 36 and thereafter, cells were given bi-weekly media changes, supplemented with 100 ng/ml CD200 and 100 ng/ml CX3CL1 to promote microglial maturation. After 2 months, microglia began to adhere to the plate and were split once by gentle washing with PBS for assays.

### Siglec quantification assays

BV-2 cells for Siglec-F quantification were assayed by plating 3k cells/well in a 24-well with 1 ml media and optional cytokines added. After 72 h incubation, cells were assayed by removing media and resuspending cells in ice-cold 100 µl flow buffer (PBS + 10% FBS) with APC-Cy7-E50-2440 (1:100) antibodies and PI (1:1,000) at 4°C with shaking for 15 min in the dark. Cells were washed with flow buffer and immediately analyzed on a BD FACS Canto flow cytometer.

iMGLs for Siglec-8 quantification were assays by plating at 2k cells/well in an 8-well glass slide with 500 µl/well of iMBM and optional cytokines and inhibitors. After 72 h, cells were fixed for 15 min at room temperature with 4% PFA and then blocked and permeabilized for an hour with 5% BSA and 0.3% Triton-X100. Cells were stained with primary antibody (ab198690: 1:100; Iba1: 1:500) overnight at 4°C. Cells were washed twice with PBS and then stained with Alexa-488/Alexa-594 secondary antibodies for 2 h. Slides were washed twice with PBS and then covered with Fluomount G solution and glass coverslip and then imaged on a Zeiss 880 confocal microscope.

For quantification of Siglec-8 IF images, we generated binary masks from Iba1 and DAPI channels. We then restricted masks to Iba1$^+$;33342$^-$ regions to generate cell masks and calculated Siglec-8 channel intensities within masks with size $> 100$ µm$^2$. Median object intensities were used for each image's quantification of Siglec-8. 2-sample *t*-test statistics on Siglec-8 intensity mean values for images were calculated using scipy (Virtanen *et al*, 2020).

### Siglec retroviral expression

Retrovirus was produced in a 10cm plate of HEK 293T cells beginning at 50–60% confluency. HEK cells were transfected for 24 h with FuGene, GAG-pol, VSVG envelope, and MSCV-IRES-Puro (MIP) plasmids expressing full length Siglec proteins with C-terminal myc epitope tags and ITIM pTyr sites mutated to phenylalanine. After a 10 ml media change and 24-h incubation, media supernatants were collected and filtered through a .45 µm filter, divided into 1 ml aliquots, and stored at −80°C.

For pINDUCER dox-inducible lentivirus expression, Siglec genes were first cloned into a pENTR gateway entry vector and then recombined into pINDUCER20 backbones using LR Clonase. Constructs were propagated in Stbl3 cells to stabilize repeat-flanked inserts. 293T cells were plated at 80% confluency and then transfected with pINDUCER transfer vectors, psPAX2, and pMD2.G at a 10:10:5 µg ratio using PEI MAX followed by a 10 ml media change after 16 h. Media supernatants were collected 72 h post-transfection, filtered through a .45 µm filter, and stored in 1 ml aliquots at −80°C.

For viral transduction, BV-2 cells were plated at 3k cells/well in a 6-well plate. For MIP-WT Siglec constructs, cells were plated at 20k cells/well due to impaired cell growth during selection. Cells were infected with virus plus 5 µg/ml polybrene for 6–8 h, followed by a fresh media change. After 24 h, cells were selected with 2 µg/ml puromycin or 1 mg/ml G418 for 72 h with daily media changes. Stable Siglec-expressing BV-2 cells were then split using 0.05% Trypsin and then centrifuged twice at 200 *g* for 3 min and washed with twice with fresh media to remove dead cells before beginning downstream assays. Inducible Siglec cells were split and frozen in 10% DMSO, 50% FBS, and 40% media. Cells were thawed and left to recover for 3–4 days. All cells were assayed post-infection within 10 days total time in culture to avoid negative selection against viral Siglec expression.

### Incucyte proliferation and endocytosis assays

BV-2 cells were plated at 3k cells/well for quantification of proliferation and endocytosis. For proliferation-only assays, treatments were added immediately on plating and cells were monitored for 1–4 days. Cells with inducible Siglec expression were co-treated with inhibitors and 500 ng/ml doxycycline. Incucyte brightfield, green-, and red-fluorescence images were collected every 2–3 h through a 10× objective. 96-wells were analyzed with 4 images per well. 24-wells were analyzed with 9 images per well.

IncuCyte ZOOM software was used to calculate cell boundaries from each brightfield image using a neural network trained on example BV-2 images. Confluency measurements were exported for the properties: Confluency = "Phase Object Confluence (Percent)", Object Count = "Phase Object Count (1/Well)", and Object Size = "Avg Phase Object Area (µm$^2$)". Confluency values were normalized to the earliest time point unobstructed by fog, bubbles, or out-of-focus cells (1–6 h, manually selected from each assay). Doubling times were calculated as $t_2 = \frac{\Delta t \ln 2}{\ln\left(\frac{y_f}{y_0}\right)}$. 2-sample *t*-test statistics on median confluency values from biological replicates were calculated using scipy.

### Cell death flow cytometry assay

To measure apoptosis and necrosis markers, BV-2 cell with stable Siglec construct expression were split into 96-well plates at a concentration of 4k cells/well with each well containing 200 µl of media and left to adhere to the plate overnight. The plate was centrifuged for 3 min at 400 *g* and 4°C, then the media was aspirated. Cells were resuspended in 50 µl ice-cold Annexin-V binding/washing buffer with Annexin-V-Alexa 488 (1:50) and Propidium Iodide (PI, 20 µg/ml) and incubated in the dark at 4°C for 15 min. After incubation, 200 µl of binding/washing buffer was added to each well. Cells were analyzed using a FACS Canto flow cytometer to determine the percentages of early apoptotic (Annexin-V$^+$), late apoptotic (Annexin-V$^+$; PI$^+$), and necrotic populations (PI$^+$).

### RNA-seq data collection

For RNA-Seq analysis of BV-2 mRNA expression, MIP-expressing cells were plated at 20k cells/well in a 6-well plate with 5 ml media. Cells were lysed after 48 h in 500 µl TRIzol and pipetted repeated through a P1000 to sheer DNA. RNA was extracted by adding 180 µl chloroform, vortexing for 15 s, and then centrifuging at 12,000 *g* for 15 min at 4°C. The top, clear, RNA-containing phase was then transferred to a new tube, mixed 1:1 with 100% Ethanol and purified using a Direct-zol RNA MicroPrep kit. RNA-seq libraries were prepared from 50 ng of RNA using Kapa mRNA Hyperprep Kits. RNA-Seq analyses were performed by The MIT BioMicro Center. Library sequencing was multiplexed on two lanes of a HiSeq 2000 (Illumina) with 40 bp reads.

mRNA libraries were prepared for 5–6 biological replicates of each Siglec receptor and control constructs. FastQ sequencing files were aligned to mm10 (ens88 annotation) with STAR and results were summarized using RSEM. Differential gene expressions were calculated using DESeq2 running under R. Single sample GSEA enrichment scores were calculated using MSigDB (h, c2cp, c5bp,

c5cc, c5mf collections) with the parameters xtools.gsea.GseaPreranked, set_min = 5, set_max = 1500, and nperm = 1000. Salmon (Patro *et al*, 2017) was used to ensure each sample had correct expression of Siglec construct sequences.

### Quantitative real-time (qRT)–PCR

qPCR analyses were performed on 20k (2xY->F) or 50k (WT Siglec) BV-2 cells plated in a 6-well plate with 5 ml media and optional inhibitors. Inducible Siglec expression cells were treated with 500 ng/ml doxycycline for 48 h. TRIzol-chloroform RNA was purified using 25 µl of Dynabeads mRNA Direct magnetic poly-T beads per sample. Beads were washed twice with 100 µl A and B wash buffers. mRNA was reverse transcribed into cDNA by adding SuperScript IV VILO Master Mix to beads and incubating in a thermocycler for 10 min at 25°C, 10 min at 50°C, and 5 min at 85°C. cDNA was then diluted 1:20 in ultrapure water and 2 µl was mixed with iQ SYBR Green Supermix and 500 nM primers for a final volume of 6 µl in a Microseal 384-Well PCR Plate. A CFX384 Touch Real-Time PCR was used for qPCR amplification and melting curves were generated to check each sample for product purity. qPCR primers are listed in Dataset EV6.

qPCR $C_q$ values were exported from the Bio-Rad CFX Manager and processed with pandas. $C_q$ values < 15 or > 40 were discarded and median $C_q$ values were calculated from 2–4 technical replicates of each sample-target combination. Fold changes were calculated as $2^{-\Delta\Delta Cq}$ for quantification cycle number relative to GADPH in each sample. 2-sample *t*-test statistics were generated using scipy.

### Endocytosis assays

For flow cytometry quantification of endocytosis, we plated 2k Siglec-expressing cells in a 96-well alongside 6k empty vector control cells in 200 µl of media. Cells were allowed to settle overnight and then fluorescent substrates were diluted in PBS and added to each well (Aβ$_{1-42}$-488: 250 ng/ml; 10 kDa Dextran-488: 250 ng/ml; 1 µm blue-green FluoSpheres: 50 million/ml). Cells were incubated with small endocytosis substrates for 3 h and larger beads for 24 h. Well media was aspirated and cells were washed once with PBS and then resuspended in 100 µl flow buffer with α-Siglec antibody (Siglec-F: APC-Cy7 E50-2440, 1:100; CD33: APC WM53, 1:100; Siglec-5: Alexa-647 194128, 1:100; Siglec-8: APC 7C9, 1:100). Cells were washed once with flow buffer and then immediately assayed. Relative fluorophore uptake was calculated as the geometric mean fluorescent intensity (gMFI) of Siglec$^+$/ Siglec$^-$ populations from each well.

To assay endocytosis of pHrodo ligands, BV-2 cells were plated and then treated the next day with inhibitors for one hour, followed by 5 µg/ml pHrodo Green Dextran (10k MW). Incucyte brightfield, and green fluorescence channels were imaged every 1 hour with a 10× objective. Object fluorescent intensities were then quantified using IncuCyte ZOOM software and exported using the metrics: "Total Object Integrated Intensity (GCU x µm$^2$/Image)".

### Siglec Co-immunoprecipitation and western blot

To perform co-IP experiments, BV-2 cells were plated in 15 cm dishes at a 1 million cells/ dish and allowed to grow until they reached 70% confluency. SHP099 was added at 1 µM 24 h prior to lysis. Cells were lysed in ice-cold protein IP buffer (1% NP-40, 150 mM NaCl, 50 mM Tris, 5 mM EDTA, pH 7.4) with HALT protease inhibitors. Lysates were spun at 13,400 rpm at 4°C for 30 min to clear lipids and DNA. Protein supernatants were transferred to a new tube and protein concentrations were quantified using a BCA assay. 2 mg of lysate was incubated with 20 µl of myc-sepharose beads overnight at 4°C. Beads were spun down at 6,500 rpm and washed three times with 1 ml protein IP buffer. Proteins were eluted by boiling beads at 95°C for 5 min in LDS Sample Buffer (4X) and 100 mM DTT.

For western blots, BV-2 cells with inducible Siglec constructs were plated in a 10-cm dish at 1 million cells/well in fresh RPMI media with 500 ng/ml doxycycline. Cells were lysed after 72 h in ice-cold RIPA with HALT protease inhibitors. Lysates were cleared, protein concentration quantified, and 50 µg aliquots were prepared by boiling with LDS and DTT as above.

Protein lysates and IP eluates were separated on a 4–12% Bis-Tris SDS–PAGE gel at 90–130 V. Proteins were then transferred to a nitrocellulose or PVDF membranes. Membranes were blocked for 1 hour at RT with Intercept Blocking Buffer and then incubated with primary antibodies: α-Siglec-F (E50-2440: 1:1,000), α-myc (9B11: 1:1,000), α-SHP-1 (C14H6: 1:1,000), α-SHP-2 (D50F2: 1:1,000), α-IL-1β (Polyclonal: 1:1,000), α-β-tubulin (1:10,000) overnight at 4°C. Membranes were washed 4 × 5 min in TBST. Membranes were incubated with Li-Cor secondary antibodies (1:20,000) for 2 h at RT and then washed 4 × 5 min in TBST and 1× in PBS. Membranes were imaged on a Li-Cor Odessey CLx.

## Data availability

All proteomics data generated in this study have been deposited on PRIDE under the accession: PXD018757 (https://www.ebi.ac.uk/pride/archive/projects/PXD018757). All sequence data have been deposited in Gene Expression Omnibus and are available under the accession: GSE149153 (https://www.ncbi.nlm.nih.gov/geo/query/acc.cgi?acc = GSE149153).

The proteomics data integration software is available on GitHub (https://github.com/white-lab/pyproteome). This repository also includes detailed software for concatenating peptide fractions on a Gibson FC 204 Fraction Collection (https://github.com/white-lab/fc-cycle), and an updated tool for validating PSMs (https://github.com/white-lab/CAMV). Sources for other code used in this study are indicated in the Reagents and Tools Table.

Detailed protocols for proteomics sample processing, phospho-peptide enrichment, C18 column preparation, and other cellular assays are available online: https://github.com/white-lab/protocols

**Expanded View** for this article is available online.

### Acknowledgements

We thank members of the White and Tsai laboratories for numerous discussions and feedback. We also thank C. Whittaker for help with RNA-seq analysis, J. Cheah for help with BV-2 assay development and screening, and R. Ahn for discussions on signal transduction mechanisms. N.M. was partially supported by the NIH Biotechnology Training Grant T32GM008334. D.M. was supported by NIRG-15-321390, and Arizona Alzheimer's Consortium. This work was supported by the Center for Precision Cancer Medicine at MIT, NIH grants U54-CA210180, R37-NS051874, RF1-AG054321, a grant from the Simons Center for the Social Brain, the Glenn Foundation and NDC Belfer (The

Neurodegeneration Consortium, The Robert A. and Renee E. Belfer Family Foundation, and the Oskar Fisher Project) to L.-H.T.

## Author contributions

NM, L-HT, and FW designed the study. AN, LAW, NM, LAA, and P-CP performed animal tissue dissections. DM collected human brain tissue samples. NM collected proteomics datasets. NM wrote proteomics data integration software. NM and BAJ designed enrichment algorithms. NM and LAA performed histology analysis. NM, WTR, and JP developed and ran iMGL assays. NM, FHR, and LL cloned DNA constructs and performed BV-2 cell assays. NM, FW, and L-HT wrote the manuscript.

## Conflict of interest

The authors declare that they have no conflict of interest.

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
