## [Review Process File · Molecular Systems Biology]

Phosphoproteomics identifies microglial Siglec-F inflammatory response during neurodegeneration

Nader Morshed, William Ralvenius, Alexi Nott, L. Watson, Felicia Rodriguez, Leyla Akay, Brian Joughin, Ping-Chieh Pao, Jay Penney, Lauren LaRocque, Diego Mastroeni, Li-Huei Tsai, and Forest White

DOI: [10.15252/msb.20209819](https://doi.org/10.15252/msb.20209819)

Corresponding author(s): Forest White (fwhite@mit.edu)

Review Timeline:

Submission Date:	27th Jun 20
Editorial Decision:	13th Aug 20
Revision Received:	15th Sep 20
Editorial Decision:	1st Oct 20
Revision Received:	2nd Oct 20
Accepted:	6th Oct 20

Editor: Maria Polychronidou

Transaction Report:

Thank you again for submitting your work to Molecular Systems Biology. We have now heard back from two of the three referees who agreed to evaluate your study. Unfortunately, after a series of reminders we have not heard back from reviewer #3. In the interest of time and since the recommendations of the other two reviewers are rather similar we have decided to proceed with making a decision based on these two available reports. As you will see below, both reviewers are overall supportive. However, they raise a series of concerns, which we would ask you to address in a revision.

I think that the recommendations of the reviewers are rather clear and there is therefore no need to repeat the points listed below. Please let me know in case you would like to discuss in further detail any of the issues raised.

On a more editorial level, we would ask you to address the following issues.

REFEREE REPORTS

Reviewer #1:

The authors take three different mouse models of Alzheimer's disease and perform phosphoproteomics on them, with special emphasis on tyrosine phosphorylation (pY). The experiments are done with a special protocol for pY enrichment that the White laboratory has developed over the years. The MS analysis is relatively conventional, using Orbitrap mass spectrometers. The analysis uses Proteome Discoverer, Mascot but then the results are analyzed once again with custom scripts and in some cases by manual interpretation, for instance of the quantification in TMT. These steps are described in the Methods section but still seem somewhat unconventional or subjective (although the authors make all scripts available). Maybe the authors can comment more on this aspect in the text and show that it is not subjective. Overall, however, the proteomics analysis appears very solid to me.

Given the scope of the journal, I would have expected some more systems biology level interpretation of the results. As it is, the authors go very quickly from the MS results to the involvement of Siglec receptors, which accounts for the vast majority of the paper. This part contains a lot of analyses and given the current interest in microglia, microglia subsets and their involvement in neurodegeneration, should be of broad interest. The fact that Siglec phosphorylation goes up in all three models is a strong indication of a central role. However, the fact that there is no direct homolog in humans of the mouse one, makes direct functional interpretation a bit difficult, although the authors certainly make a tremendous effort and present many interesting results. All in all, I find this work to be very interesting and solid and a good match to the journal. As mentioned above, the structure could perhaps be improved somewhat to balance the story and make it flow better. A Cell style final cartoon of the proposed mechanism may help to put it all together in the end.

Minor points:

- the authors could eliminate some jargon and abbreviations that may not be so easy to understand for the broad readership of the journal (example, 'm.o. mice').

- From Figure 1, it is not immediately clear where the control mouse channel goes and how samples are paired into TMT sets (different brain regions, mouse models and replicate animals)
- Throughout the manuscript it would be good to get something more than upregulated or downregulated, i.e. p-values and effect sizes.
- Some data availability refers to nature.com; presumably from another submission (p.39)

Reviewer #2:

Signaling pathways underlying the development of Alzheimer's disease (AD) may involve multiple toxic insults (e.g. Aβeta, tau, inflammation, kinase deregulation, etc.) and diverse cell types/brain regions during different disease stages. Although genetic studies reveal some causative/ risk genes, the investigation of the related pathways has been significantly delayed. In this manuscript, two expert groups have collaborated to use the latest phosphoproteomic strategy to probe the activation of signaling pathways in several commonly used AD mouse models (CK-p25, 5XFAD, and Tau P301S), analyzing 10,828, 24,365, 11,853 unique phosphopeptides in these animals, respectively. The 5XFAD/Δp35KI mice were also analyzed to study the role of p25CDK5 in the 5x FAD background. Dr. White is a pioneer in the development of phosphoproteomics, and in this study, his group uniquely highlighted the analysis of Tyr phosphorylation events, which is often underrepresented in other global phosphoproteomic projects. Cross-model comparisons led to the identification of shared kinases and microglial activation. Then the authors focused on the functional studies of activated Siglec-F (a CD33-related Siglec gene family member) in microglia in a murine microglial cell line (BV-2) and human stem-cell derived microglia. While Siglec-F was upregulated upon interferon gamma-induced microglial activation, Siglec-F overexpression activates an endocytic and cell death-related inflammatory pathway, dependent on its sialic acid substrates and ITIM motif phosphosites. They also showed that the human homolog Siglec-8 was upregulated in human AD brain samples. Overall, the proteomics study was comprehensive and carried out with sufficient replicates, followed by extensive hypothesis-driven experiments to study the role of Siglec-F in microglial activation, although the experiments were largely limited to cellular models. The topic of this manuscript is important and the results might be highly relevant to AD pathogenesis. The manuscript is suitable for publication if some concerns are addressed.

1. The initial part of the manuscript is an excellent resource of phosphoproteome in AD mouse models. It is expected that all raw MS data should be released to a public database (e.g. <https://www.ebi.ac.uk/pride/archive>)
2. The sample information is not clearly presented in TABLE_EV1-suppl. In the worksheet of "Mouse Tissue", 53 samples are listed. In the worksheet of "MS Runs", 83 lanes are shown with pY and pST analysis. In the worksheet of results (e.g. 5x FAD), the title of each column is not well explained, such as "7387 5XFAD Δp35KI 9mo Hip". It is not obvious how the samples were distributed in different TMT batches, how to match each channel (e.g. 126, 127N, etc.) in each TMT batch with the 53 samples, and how to normalize batch effects. A single summary table of samples, MS runs, and MS results, would be more informative.
3. Because of the undersampling issue, comparable phosphopeptides in different TMT batches were reduced. When merging results from different TMT batches, missing values often occur. It is not clear how the authors set up cutoffs for the missing values. How many phosphopeptides were constantly detected and compared in all samples? Although this limitation may be unavoidable due to restrictions of the current MS method, the problem is gradually overcome by further development in the proteomics field. This point needs to be thoroughly discussed and clarified.
4. It appears that the authors used the TMT-MS2 method for quantification, in which the accuracy

of quantification may be affected by ratio compression. It should be discussed how the authors reduced the impact of ratio compression in the analysis.

5. The false discovery rate for phosphopeptide identification was based on Percolator FDR $< 5e-2$, which is acceptable and larger than regular 0.01. This important cutoff may be directly presented in the main text.

6. During the DE analysis, it is not clear how the authors set up cutoffs (e.g. fold change, p values or q value/FDR).

7. For this journal of Molecular Systems Biology, it is expected that some types of pathway/network analysis might be performed to interpret these large scale datasets.

8. The author might present more rationale on selecting the Siglec-F for downstream studies (e.g. in the CD33-related Siglec gene family). How to prioritize the identified DE phosphoproteins relevant to AD pathogenesis? The Siglec gene family has many members including CD33. Are other Siglec proteins also found to be changed in the analysis?

9. This reviewer appreciates the thorough investigation of Siglec-F function in the cellular models. One concern is that the cellular model experiments generated largely correlative data. What is the effect of Siglec-F knockout in the cells during interferon gamma stimulation? Do multiple Siglec family proteins play a redundant role in the pathway?

Summary of Responses:

We thank the reviewers for their careful and thoughtful comments. We have provided a second version of the manuscript with the following changes:

Reviewer #1:

The authors take three different mouse models of Alzheimer's disease and perform phosphoproteomics on them, with special emphasis on tyrosine phosphorylation (pY). The experiments are done with a special protocol for pY enrichment that the White laboratory has developed over the years. The MS analysis is relatively conventional, using Orbitrap mass spectrometers. The analysis uses Proteome Discoverer, Mascot but then the results are analyzed once again with custom scripts and in some cases by

manual interpretation, for instance of the quantification in TMT. These steps are described in the Methods section but still seem somewhat unconventional or subjective (although the authors make all scripts available). Maybe the authors can comment more on this aspect in the text and show that it is not subjective. Overall, however, the proteomics analysis appears very solid to me.

Given the scope of the journal, I would have expected some more systems biology level interpretation of the results. As it is, the authors go very quickly from the MS results to the involvement of Siglec receptors, which accounts for the vast majority of the paper. This part contains a lot of analyses and given the current interest in microglia, microglia subsets and their involvement in neurodegeneration, should be of broad interest. The fact that Siglec phosphorylation goes up in all three models is a strong indication of a central role. However, the fact that there is no direct homolog in humans of the mouse one, makes direct functional interpretation a bit difficult, although the authors certainly make a tremendous effort and present many interesting results.

All in all, I find this work to be very interesting and solid and a good match to the journal. As mentioned above, the structure could perhaps be improved somewhat to balance the story and make it flow better. A Cell style final cartoon of the proposed mechanism may help to put it all together in the end.

Response: We thank the reviewer for these thoughtful comments, and have tried to address their concerns. To clarify the steps involved in quantification of TMT: We use an open-source set of scripts that read peptide-spectrum matches (PSMs) and TMT quantification values directly from Proteome Discoverer .msf files into a pandas DataFrame for analysis in python. Due to our interest in Siglec-F, we manually validated and quantified 5 scans of SVyTEIK (Siglec-F pY561) and inserted these values into the DataFrame for its respective MS analysis. These scans all matched the MS2 pattern that is shown in Figure EV3E, had m/z values of 689.37, a charge state of +2, and retention times of 55-85 min. For clarity, we have listed all scans that were inserted after manual validation:

MS Analysis	Raw File	MS2 Scan	Peptide
CK-7wk-H1-pY	2019-04-24-CKp25-SiglecF-1-py-SpinCol-col189.raw	23074	SV(pY)TEIK
CK-7wk-H2-pY	2019-04-24-CKp25-SiglecF-2-py-SpinCol-col181.raw	10020	SV(pY)TEIK
p35-X1-pY	2016-12-06-p35X1-pY-HighSelect-pre63-col88-1.raw	17015	SV(pY)TEIK
FAD-X4-pY	2017-02-16-FADX4-pY-imac52-pre101-col101.raw	12742	SV(pY)TEIK
Tau-6moHR4-pY	2018-06-26-Tau6moHR4-pY-imac-pre141-col151.raw	14403	SV(pY)TEIK

To quantify TMT abundances in these scans, we manually measured the TMT peak intensities using Xcalibur Qual Browser. These values varied from the exact values calculated by Proteome Discoverer's Reporter Ion "Most Confident Centroid" method. We compared TMT quantification values for scans that we matched by Proteome

Discoverer and observed <5% difference between the two methods, indicating that manual TMT quantification is generally accurate.

Our downstream processing workflow to filter, normalize, and integrate PSMs was designed specifically for this study. The filtering parameters for MASCOT Ion Score, Percolator FDR, and Proteome Discoverer Isolation Interference cutoffs were taken from previous projects in our lab. In an independent, unpublished analysis we found low correlation between TMT abundances for identical peptides that had a median TMT quantification value below 1500. We therefore included this cutoff to improve quantification accuracy.

To normalize our dataset, we used a method derived from the CONSTrained STANdardization (CONSTANd) algorithm, referenced in the text. This procedure iteratively normalizes the TMT values for each PSM to their mean TMT intensities. It then normalizes each TMT channel to its mode TMT intensity (estimated with a KDE function). This corrects for differences in peptide loading input amounts and generates a matrix of TMT quantification values that is centered around 1.

To integrate TMT quantification values, we combine TMT quantification information for PSMs that have identical peptide sequence and phosphorylation / oxidation modifications. To calculate the integrated abundances, we used a weighted mean function which biases quantification values towards peptides with high TMT intensities and low isolation interference. Finally, we normalize quantification values to the median of the control animal / cell line group.

While most of this was contained in an abbreviated manner in the Methods, we added the following sentence to alleviate concerns regarding manual quantification: “We compared TMT quantification values for scans that we matched by Proteome Discoverer and observed <5% difference between the two methods, indicating that manual TMT quantification is generally accurate.”

With regard to systems analysis, this point was a concern of Reviewer #2 as well. To address this concern, we performed multiple additional analyses of our data sets, including using PHOTON for clustering and protein-protein interaction networks. Unfortunately, clustering did not find any interesting peptide trends that were not captured by a simple group comparison and worked poorly for cross model comparisons due to the sparseness of the matrix (due to data-dependent analysis). PHOTON gave a number of predicted protein-protein interactions, but did not provide consistent changes between its predictions and the phosphosites that we directly measured in each model:

In the end we found that gene ontology enrichment was most reflective of the actual peptide trends that we identified. These new figures have been added to Figure EV3A-EV3C. We have also added text to results section under “Cross-model comparison identifies shared kinase and microglial activation” and methods section under “Phosphosite enrichment and pathway analysis” discussing pathway analysis using gene ontology enrichment.

Minor points:

- the authors could eliminate some jargon and abbreviations that may not be so easy to understand for the broad readership of the journal (example, 'm.o. mice').

Response: We have attempted to remove the use of jargon in the second version of the manuscript.

- From Figure 1, it is not immediately clear where the control mouse channel goes and how samples are paired into TMT sets (different brain regions, mouse models and replicate animals)

Response: We have clarified this point by listing all sample -> TMT channel mapping for all mouse and cell line samples in **Dataset EV1** and **Dataset EV5**.

- Throughout the manuscript it would be good to get something more than upregulated or downregulated, i.e. p-values and effect sizes.

Response: We have added p-values and effect sizes to the text of the manuscript wherever upregulation or downregulation are mentioned.

- Some data availability refers to nature.com; presumably from another submission (p.39)

Response: Thank you for catching this mistake, we have removed it in the second version of the manuscript.

Reviewer #2:

Signaling pathways underlying the development of Alzheimer's disease (AD) may involve multiple toxic insults (e.g. Abeta, tau, inflammation, kinase deregulation, etc.) and diverse cell types/brain regions during different disease stages. Although genetic studies reveal some causative/ risk genes, the investigation of the related pathways has been significantly delayed. In this manuscript, two expert groups have collaborated to use the latest phosphoproteomic strategy to probe the activation of signaling pathways in several commonly used AD mouse models (CK-p25, 5XFAD, and Tau P301S), analyzing 10,828, 24,365, 11,853 unique phosphopeptides in these animals, respectively. The 5XFAD/ Δ p35KI mice were also analyzed to study the role of p25CDK5 in the 5xHAD background. Dr. White is a pioneer in the development of phosphoproteomics, and in this study, his group uniquely highlighted the analysis of Tyr phosphorylation events, which is often underrepresented in other global phosphoproteomic projects. Cross-model comparisons led to the identification of shared kinases and microglial activation. Then the authors focused on the functional studies of activated Siglec-F (a CD33-related Siglec gene family member) in microglia in a murine microglial cell line (BV-2) and human stem-cell derived microglia. While Siglec-F was upregulated upon interferon gamma-induced microglial activation, Siglec-F overexpression activates an endocytic and cell death-related inflammatory pathway, dependent on its sialic acid substrates and ITIM motif phosphosites. They also showed that the human homolog Siglec-8 was upregulated in human AD brain samples. Overall, the proteomics study was comprehensive and carried out with sufficient replicates, followed by extensive hypothesis-driven experiments to study the role of Siglec-F in microglial activation, although the experiments were largely limited to cellular models. The topic of this manuscript is important and the results might be highly relevant to AD pathogenesis. The manuscript is suitable for publication if some concerns are addressed.

1. The initial part of the manuscript is an excellent resource of phosphoproteome in AD mouse models. It is expected that all raw MS data should be released to a public database (e.g. <https://www.ebi.ac.uk/pride/archive>)

Response: All proteomics data generated in this study has been deposited on PRIDE under the accession: PXD018757. This deposition includes raw MS data and searched ProteomeDiscoverer 2.2 .msf files. We mistakenly omitted the reviewer account login

info from the initial submission and have added it to the second version of the manuscript.

2. The sample information is not clearly presented in TABLE_EV1-suppl. In the worksheet of "Mouse Tissue", 53 samples are listed. In the worksheet of "MS Runs", 83 lanes are shown with pY and pST analysis. In the worksheet of results (e.g. 5xFAD), the title of each column is not well explained, such as "7387 5XFAD Δ p35KI 9mo Hip". It is not obvious how the samples were distributed in different TMT batches, how to match each channel (e.g. 126, 127N, etc.) in each TMT batch with the 53 samples, and how to normalize batch effects. A single summary table of samples, MS runs, and MS results, would be more informative.

Response: We have clarified the summary table of MS samples, runs, and results in the newest version of Dataset EV1 and Dataset EV5. The batch processing and normalization procedure is described in our methods section as well as in our reply to Reviewer #1.

3. Because of the undersampling issue, comparable phosphopeptides in different TMT batches were reduced. When merging results from different TMT batches, missing values often occur. It is not clear how the authors set up cutoffs for the missing values. How many phosphopeptides were constantly detected and compared in all samples? Although this limitation may be unavoidable due to restrictions of the current MS method, the problem is gradually overcome by further development in the proteomics field. This point needs to be thoroughly discussed and clarified.

Response: This comment highlights an important point about our data. Because we used a data-dependent MS acquisition method, many peptides were not identified in every MS analysis. The overlap between unique peptides identified in technical replicates (separate TMT 6/10-plex labeled experiments for the same proteolytic digests) was 45-70% for hippocampus and cortex samples from CK-p25 and 5XFAD mice. The overlap between peptides identified between hippocampus and cortex tissue from the same set of mice was 68% for CK-p25 and 64% for 5XFAD. And the overlap between peptides identified from different animal models was 60% between CK-p25 and 5XFAD. Out of 30,370 unique peptides that were identified in any MS analysis, 12% (305 pY, 2900 pST, 3684 total peptides) were identified in at least one comparison group for each of the three animal models.

Because data-dependent mass spectrometry has well-characterized reproducibility limitations, we ran technical replicates to increase the total number of identified peptides. We then combined TMT quantification data from all runs into one normalized peptide abundance matrix. To identify trends, we compare TMT quantification values for samples within the same set of TMT 6/10-plex analyses.

We have avoided any conclusions that would be derived from an absence of MS evidence (i.e. If Gfap pY321 is only identified in CK-p25 hippocampus, this does not imply that Gfap does not change in any other tissue or mouse model).

We considered adding a discussion of these points to the manuscript, but we thought that this addition would negatively affect the flow of the manuscript.

4. It appears that the authors used the TMT-MS2 method for quantification, in which the accuracy of quantification may be affected by ratio compression. It should be discussed how the authors reduced the impact of ratio compression in the analysis.

Response: TMT ratio compression is indeed a problem that can affect quantification accuracy and suppress large changes in proteomics analyses. We have reduced the effects of ratio compression in our data by (1) simplifying the sample by enriching for low-abundance phosphopeptides to remove background proteome signal, (2) using a narrow precursor mass isolation window of 0.4 Da (total width), and (3) weighting against peptide isolation interference when combining quantification data from peptide-spectrum matches for identical peptides. We added the following sentence to the Methods section:

“Isolation width was set to 0.4 Da (full width) to decrease isolation interference and TMT ratio compression.”

Using this approach, we were able to identify fold changes between diseased and control animals as large as 8.9x (CK-p25: Cdk1/2/3 pT14), 4.2x (5XFAD: Siglec5 pY561), and 10.4x (Tau P301S: MAPT pT152).

5. The false discovery rate for phosphopeptide identification was based on Percolator FDR < 5e-2, which is acceptable and larger than regular 0.01. This important cutoff may be directly presented in the main text.

Response: We have re-run our analysis with a Percolator FDR cutoff of 0.01 and observed only minor changes in the generated figures. The second version of the manuscript uses an FDR cutoff of 0.01.

6. During the DE analysis, it is not clear how the authors set up cutoffs (e.g. fold change, p values or q value/FDR).

Response: For the Venn diagrams and heatmaps, we used a fold change cutoff of 1.25x and 2-sample t-test p-value cutoff of 0.01 to identify peptides with differential abundance. We have clarified this point in the text.

7. For this journal of Molecular Systems Biology, it is expected that some types of pathway/network analysis might be performed to interpret these large scale datasets.

Response: See response to Reviewer #1, above. Briefly, after several analysis attempts, we settled on gene-ontology enrichment and have added analysis figures to Figure EV3A-C and changed the text in “Cross-model comparison identifies shared kinase and microglial activation” to add an additional systems analysis.

8. The author might present more rationale on selecting the Siglec-F for downstream studies (e.g. in the CD33-related Siglec gene family). How to prioritize the identified DE phosphoproteins relevant to AD pathogenesis? The Siglec gene family has many members including CD33. Are other Siglec proteins also found to be changed in the analysis?

Response: We identified Siglec-F pY561 as being one of two phosphosites that were consistently upregulated across all diseased models and tissues. We selected Siglec-F for follow-up validation as it is a cell surface receptor that was predicted to be expressed on microglia and little was known at the time about its role in Alzheimer’s disease. In our follow-up analysis of available single-cell RNA-seq data of microglia from CK-p25 and 5XFAD mice, we observed that Siglec-1, Cd22, and Siglec-F appear to be upregulated at the transcript level in Lpl⁺ / Spp1⁺ late-response inflammatory microglia (**Figure EV5A**). In 5XFAD animals, Cd22, and Siglec-F are upregulated at the transcript level in the Lpl⁺ / Spp1⁺ disease-associated microglia population (**Figure EV5B**).

9. This reviewer appreciates the thorough investigation of Siglec-F function in the cellular models. One concern is that the cellular model experiments generated largely correlative data. What is the effect of Siglec-F knockout in the cells during interferon gamma stimulation? Do multiple Siglec family proteins play a redundant role in the pathway?

Response: While we do see reduced growth rates in BV-2 cells that are treated with interferon gamma, we do not know if this is mediated by Siglec-F. Due to time constraints and COVID-19 lab shutdowns, we did not test the effects of Siglec-F / Siglec-8 knockout in the context of interferon gamma stimulation.

With regards to Siglec receptor redundancy, in mice, it appears that only Cd22 and Siglec-F are upregulated in the late response inflammatory microglia seen in a scRNA-seq analysis of CK-p25 mice. Cd22 and Siglec-F also appear to be upregulated in the disease-associated microglia population identified in 5XFAD mice. While other Siglecs change in other microglial populations, Cd22 and Siglec-F appear to be the relevant Siglecs to this population. We focus on Siglec-F as *Pluvinage et al. (2019)* have shown that Cd22 has functional effects on microglia during aging.

In humans, the expression patterns may be more complicated. We have identified that Siglec-8 is upregulated on microglia in late-onset AD. *Griciuc et al. (2013)* have identified that CD33 is upregulated on microglia in AD. Due to the low coverage of reads in the available single-cell/single-nuclei RNA-seq datasets of human microglia, we have not been able to answer whether other Siglecs are also differentially regulated in AD microglia.

One additional point is that within the Siglec family, Siglec-8 and Siglec-F both uniquely bind 6'-sulfo Sialyl Lewis X. Siglec-F has a slightly more permissive binding pattern that also includes Sialyl Lewis X. This suggests that Siglec-F and Siglec-8 bind to similar substrates in the brain, although the more permissive binding of Siglec-F may indicate that it plays a more general role in mice compared to Siglec-8 in humans.

2nd Editorial Decision**1st Oct 2020**

Thank you for sending us your revised manuscript. We think that the performed revisions satisfactorily address the issues raised by the reviewers. I am glad to inform you that we can soon formally accept your manuscript for publication, pending some editorial issues listed below.

2nd Authors' Response to Reviewers**2nd Oct 2020**

The Authors have made the requested editorial changes.

Accepted**6th Oct 2020**

Thank you again for sending us your revised manuscript. We are now satisfied with the modifications made and I am pleased to inform you that your paper has been accepted for publication.

Corresponding Author Name: Forest M. White
 Journal Submitted to: Molecular Systems Biology
 Manuscript Number: MSB-20-9819